# Potential decoupling of $CO_2$ and Hg uptake process by global vegetation in the 21st century

Tengfei Yuan[1], Shaojian Huang[1], Peng Zhang[1], Zhengcheng Song[1,2,3], Jun Ge [1,3], Xin Miao [1], Yujuan Wang [1], Qiaotong Pang[1], Dong Peng[1], Peipei Wu[1], Junjiong Shao [4], Peipei Zhang[5], Yabo Wang[6], Hongyan Guo[7], Weidong Guo [1] & Yanxu Zhang [1,2,3] ✉

Mercury (Hg), a potent neurotoxin posing risks to human health, is cycled through vegetation uptake, which is susceptible to climate change impacts. However, the extent and pattern of these impacts are largely unknown, obstructing predictions of Hg's fate in terrestrial ecosystems. Here, we evaluate the effects of climate change on vegetation elemental Hg [Hg(0)] uptake using a state-of-the-art global terrestrial Hg model (CLM5-Hg) that incorporates plant physiology. In a business-as-usual scenario, the terrestrial Hg(0) sink is predicted to decrease by 1870 Mg yr$^{-1}$ in 2100, that is ~60% lower than the present-day condition. We find a potential decoupling between the trends of $CO_2$ assimilation and Hg(0) uptake process by vegetation in the 21st century, caused by the decreased stomatal conductance with increasing $CO_2$. This implies a substantial influx of Hg into aquatic ecosystems, posing an elevated threat that warrants consideration during the evaluation of the effectiveness of the Minamata Convention.

Mercury (Hg) is a pervasive toxic pollutant causing adverse effects on human health at a global scale[1]. Additionally, it endangers ecosystems by bioaccumulating in food chains, affecting biodiversity and disrupting ecological balances[2]. Anthropogenic activities, such as fossil fuel combustion and metal mining, have significantly increased Hg emissions and caused widespread environmental Hg contamination since the industrial era[3,4]. Vegetation within terrestrial ecosystems can absorb large amounts of atmospheric gaseous elementary Hg [Hg(0)] (2200–3600 Mg year$^{-1}$), acting as a major sink for the atmosphere in the present-day Hg cycles[5,6]. Long-living vegetation not only stores present-day Hg but also Hg emitted into the atmosphere decades ago[7].

Climate-related factors, such as rising temperatures and elevated carbon dioxide (e$CO_2$), are profoundly affecting the growth and physiological processes of vegetation[8,9]. However, their cascading effects on the terrestrial Hg cycle, especially the uptake of Hg(0) by vegetation, remain unclear. Here, we evaluate this effect by using a coupled climate–land–mercury model running for the twenty-first century.

Previous research has revealed complex impact pathways of climate change on vegetation Hg(0) uptake. The higher temperature was found to boost vegetation's ability to absorb atmospheric Hg(0) in glacier retreat areas[10]. Yet, the rising temperatures often lead to localized droughts, which are likely to weaken the Hg sink in terrestrial

¹School of Atmospheric Sciences, Nanjing University, Nanjing, Jiangsu, China. ²Frontiers Science Center for Critical Earth Material Cycling, Nanjing University, Nanjing, Jiangsu, China. ³Joint International Research Laboratory of Atmospheric and Earth System Sciences, Nanjing University, Nanjing, Nanjing, Jiangsu, China. ⁴State Key Laboratory of Subtropical Silviculture, College of Forestry and Biotechnology, Zhejiang A&F University, Hangzhou, China. ⁵CAS Key Laboratory of Mountain Ecological Restoration and Bioresource Utilization & Ecological Restoration and Biodiversity Conservation Key Laboratory of Sichuan Province, Chengdu Institute of Biology, Chinese Academy of Sciences, Chengdu, China. ⁶College of Environmental Science and Engineering, Yangzhou University, Yangzhou, China. ⁷State Key Laboratory of Pollution Control and Resource Reuse, School of the Environment, Nanjing University, Nanjing, China. ✉e-mail: zhangyx@nju.edu.cn

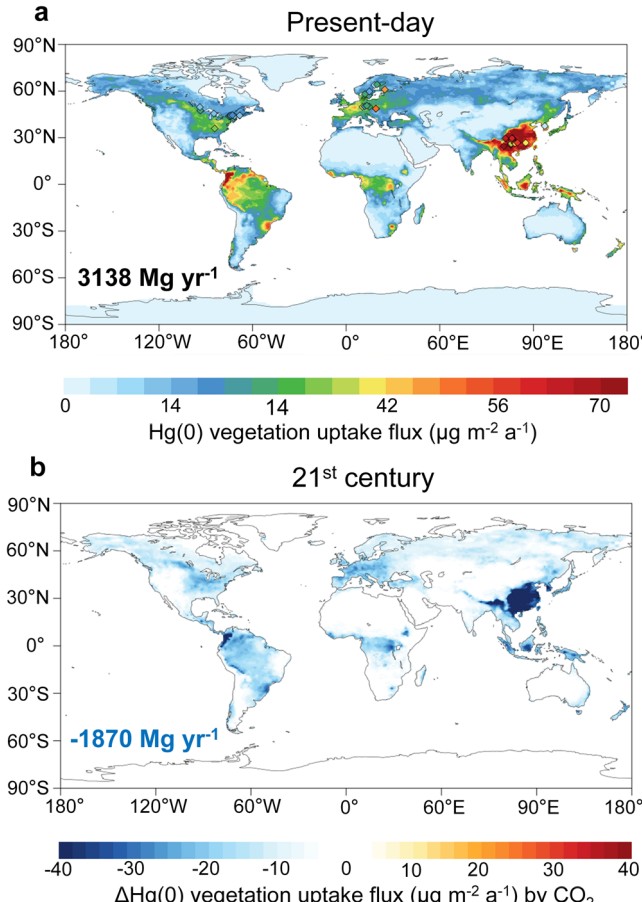

**Fig. 1 | The biogeochemical effect of elevated carbon dioxide on global vegetation uptake of Hg(0). a** Hg(0) vegetation uptake flux at present day, the value represents the gross uptake of atmospheric Hg(0) through both stomatal and cuticular (non-stomatal) processes and does not include the immediate re-emission from foliage. Observations (represented by rhombuses) are obtained from the global vegetation measurements database (see Materials and Methods). **b** Change in Hg(0) vegetation uptake flux caused by the biogeochemical effects of $eCO_2$ between 2100 and the present day under a business-as-usual scenario (SSP-8.5). The numbers in the figure represent the global total values.

ecosystems[11]. Furthermore, the alteration in global precipitation patterns can affect the Hg sink of forests[12]. The future increases in vegetation density driven by $CO_2$ fertilization are expected to enhance the Hg(0) dry deposition velocity[13,14]. Stomatal uptake of Hg(0) by foliage was found to be inhibited under high vapor pressure deficit (VPD) conditions and proved to be sensitive to extreme climate events[15–17]. However, previous studies have not systematically considered the interaction of climate change, vegetation dynamics, and Hg processes, and some have primarily focused only on specific regions[14,18].

Therefore, our research aims to exam how vegetation-regulated atmospheric Hg(0) deposition will change under the impact of future climate change. We hypothesize that the climate will influence global vegetation Hg(0) uptake by altering the plant physiology such as the stomatal activities, with $CO_2$ and other meteorological factors as important driver factors. This study uses the Community Land Model–Hg (CLM5-Hg) within the Community Earth System Model (CESM), which incorporates a dynamic plant growth framework that includes the stomatal uptake process of Hg(0) and various plant functional types (PFTs)[19]. This model also encompasses the comprehensive biogeochemical cycling of Hg in terrestrial ecosystems. It is forced by different future climate scenarios throughout the twenty-first century: (i) the Shared Socioeconomic Pathway (SSP) 1-2.6, a.k.a. the

the "2 °C scenario" representing a sustainability framework; (ii) the SSP3-7.0, representing a medium-high reference within the socioeconomic context of "regional rivalry"; (iii) SSP5-8.5, a.k.a. the "business-as-usual", considered as the worst-case scenario within a high fossil fuel-intensive world[20]. These future scenarios are compared against the present-day simulation (baseline case). We also include a pre-industrial scenario (ca. 1850) for comparison. Furthermore, we design sensitivity experiments by alternatively changing specific climate-related factors for the SSP5-8.5 scenario while maintaining others consistent with the baseline scenario (Supplementary Table 1). These experiments can diagnose and compare the influence of individual factors on the uptake of Hg(0) by vegetation in terrestrial ecosystems. We consider factors including atmospheric $CO_2$ concentration (with a focus on biogeochemical effects only), precipitation, temperature, humidity, pressure, radiation, and wind. We keep the anthropogenic Hg emissions and the atmospheric Hg concentrations constant for all scenarios to highlight the impact of climate factors, and remove the effects of land use and land cover change (LULCC) and aerosols (see "Methods").

## Results and discussion
### Reduced Hg(0) uptake
Our results indicated that, under future climate change scenarios, the biogeochemical effects of elevated $CO_2$ emerge as the dominant factor influencing vegetation Hg(0) uptake. This uptake represents the gross uptake of atmospheric Hg(0) through both stomatal and cuticular (non-stomatal) processes and does not include the immediate re-emission from foliage. In the SSP5-8.5 (business-as-usual) scenario where only $CO_2$ concentration is altered and other climatological factors kept as present day, the global Hg(0) uptake decreases by 1870 Mg year$^{-1}$ or 59.6% in 2100, in comparison to the present-day condition of 3138 Mg year$^{-1}$ (Fig. 1). The most significant changes were simulated in East Asian and the Amazon forests, attributed to their high Hg(0) assimilation compared to other regions[19]. The global vegetation Hg(0) uptake was predicted to further decrease by only 88 Mg year$^{-1}$ while accounting for the changes of all factors in the SSP5-8.5 scenarios (Supplementary Fig. 2). Other climate change factors, such as changing temperature, precipitation, radiation, pressure, and humidity account for a much smaller effect than the biogeochemical effects of $eCO_2$ alone (Supplementary Figs. 3 and 4b). We also found no significant interaction between the $eCO_2$ effect and other factors (Supplementary Fig. 5).

We predicted a higher global vegetation Hg(0) uptake for the other two future scenarios with lower $CO_2$ levels: 2685 Mg year$^{-1}$ for the SSP1-2.6 (2 °C) scenario ($CO_2$ = 445.6 p.p.m.) and 1570 Mg year$^{-1}$ for SSP3-7.0 (regional rivalry) scenario ($CO_2$ = 867.2 p.p.m.) vs. 1268 Mg year$^{-1}$ for SSP5-8.5 (business-as-usual) scenario ($CO_2$ = 1135.2 p.p.m.) (Fig. 2). A slight increase in Hg(0) uptake was simulated during the pre-industrial era: 3324 Mg year$^{-1}$ when the global average $CO_2$ is lower at 288 p.p.m. Unlike the future scenarios, we noted the largest impact is contributed by the lower atmospheric humidity in the pre-industrial era (Supplementary Figs. 3d and 4a). Changes in precipitation and temperature, as well as its interaction with the biogeochemical effects of $eCO_2$, significantly affect the uptake of Hg(0) by global vegetation (see Supplementary Figs. 4a and 5a). This suggested that the biogeochemical impact of $eCO_2$ on Hg(0) uptake has not yet become dominant when compared with other climate change factors in the pre-industrial era. When all scenarios were considered together, we observe a continuous decrease in the potential of vegetation to uptake Hg(0) in the future as $CO_2$ levels increase (Supplementary Fig. 12).

Direct field evidence that integrated the various processes of terrestrial Hg cycling, and was responsive to multiple climate change factors across different spatiotemporal scales, remained scarce. Previous manipulative experiments focusing on $CO_2$ enrichment have primarily concentrated on the biogeochemical impact of $eCO_2$ on

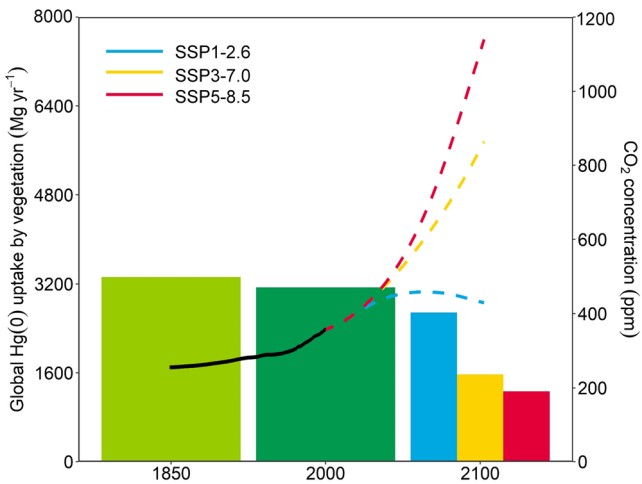

**Fig. 2 | Global vegetation Hg(0) uptake (bars) via elevated $CO_2$ under different atmospheric $CO_2$ concentrations (lines) from the historical emission scenario in 1850 to the Shared Socioeconomic Pathway (SSP) emission scenarios in 2100.** SSP1-2.6 represents the lowest scenario, termed the "2 °C scenario," which aims for a sustainable future. SSP3-7.0 represents a moderate scenario, described as a medium-high reference scenario within the socio-economic context of "regional rivalry." SSP5-8.5 represents the highest scenario, also known as "business-as-usual," considered the worst-case scenario in a high fossil fuel-intensive world. Solid lines represent atmospheric $CO_2$ levels during 1850–2000, shaded lines represent the atmospheric $CO_2$ levels during 2000–2100 under different scenarios.

vegetation Hg concentrations (Fig. 3a). The $CO_2$ concentrations were usually enhanced to 360–610 p.p.m. in these experiments, similar to our future scenarios. Data on $Hg_{veg}$ (Hg flux or Hg concentration of plant) from six $eCO_2$ experimental studies were integrated into four $eCO_2$ conditions based on the levels of increased $CO_2$ concentration (Supplementary Fig. 10). Under experimental conditions of a 150 and 200 p.p.m. increase in $CO_2$ concentration, $Hg_{veg}$ showed a significant decreasing trend ($P < 0.05$, Supplementary Fig. 10a). Despite the lack of significant differences at 253 and 360 p.p.m., even showing an opposite trend at 253 p.p.m. (Supplementary Fig. 10b), the overall effect of our meta-analysis ($P < 0.01$) suggested that $eCO_2$ had a suppressive effect on vegetation Hg levels (Fig. 3b). The meta-analysis revealed a significant decrease in vegetation Hg levels as a result of $eCO_2$, showing an average decrease in foliage Hg levels or Hg uptake of 5.87% per 100 p.p.m. increase (95% CI, −6.5% to −5.3%). If translated to the increased $CO_2$ concentration level under our model's SSP5-8.5 scenario, this change rate reached nearly 50%. The change in terrestrial Hg(0) sink (-60%) simulated by our model was fairly close to the experimentally observed values. Overall, our findings are consistent with existing evidence synthesized from experimental data worldwide, as atmospheric uptake is the major source of Hg in foliage[6,21]. Indeed, a multitude of studies, including those employing isotopic techniques, have demonstrated that a predominant portion of atmospheric Hg(0) is assimilated into terrestrial ecosystems via vegetation, primarily through stomatal uptake, accounting for over 80% of the total uptake[22,23].

We found that this effect varies across different PFTs. The most substantial suppression by $eCO_2$ was observed in crops (−12.2 ± 1.1% per 100 p.p.m. $CO_2$), while the suppression in grasses (−5.8 ± 1.1% per

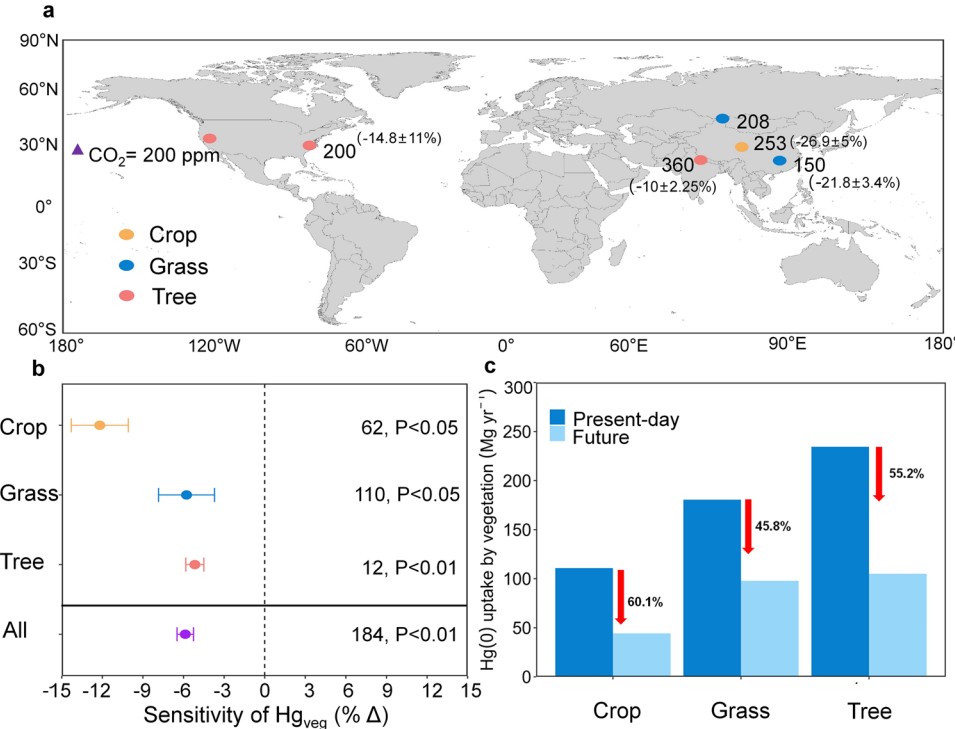

**Fig. 3 | The biogeochemical effect of $eCO_2$ on global vegetation Hg(0) uptake.**
**a** Global distribution of $eCO_2$ experiments included in this meta-analysis. Circles of orange, blue, and red indicate experiments on crops, grass, and tree, respectively. The numbers outside the parentheses represent the increase in $CO_2$ concentration, while the numbers inside the parentheses indicate the corresponding change rate in $Hg_{veg}$ (based on the four integrated Δ $CO_2$ levels). **b** The sensitivity of vegetation

Hg in response to $eCO_2$ in different types of plants across different experimental studies. The triangle represents one unit of $eCO_2$ (100 p.p.m. increase). Each data point represents the weighted mean values; error bars indicate the 95% confidence intervals. The numbers indicate sample sizes with the number of stars representing significant levels. **c** Modeled annual global Hg(0) uptake by different types of vegetation under present day and future $CO_2$ levels under the SSP5-8.5 scenario.

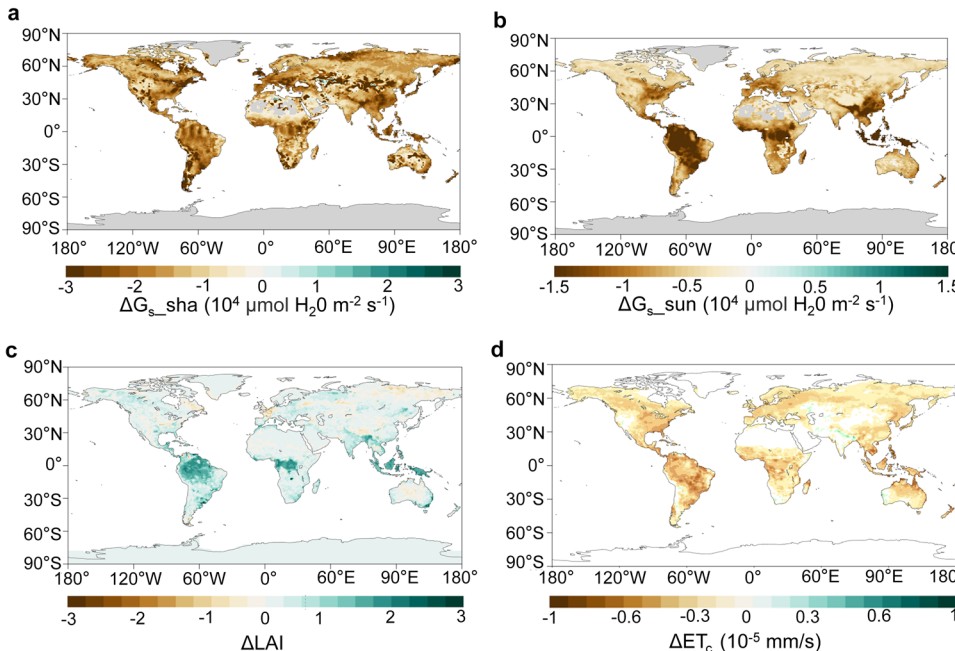

**Fig. 4 | The biogeochemical effect of eCO₂ on stomatal conductance and leaf area index (LAI) under the SSP5-8.5 scenario. a** Changes in global sunlit stomatal conductance ($\Delta G_s$_sun) caused by eCO₂ between present day and 2100. **b** Changes in global shaded stomatal conductance ($\Delta G_s$_sha) caused by the eCO₂ between present day and 2100. **c** Changes in global leaf area index ($\Delta$LAI) caused by the eCO₂ between present day and 2100. **d** Changes in evapotranspiration from canopy ($\Delta$ET$_c$) caused by the eCO₂ between present day and 2100.

100 p.p.m. CO₂) and trees (−5.2 ± 0.3% per 100 p.p.m. CO₂) was relatively similar (Fig. 3b). In our model, similar pattern was simulated. We also extracted the Plant Functional Types (PFTs) corresponding to the species used in the meta-analysis experimental data. We found that the modeled global Hg(0) uptake of trees, crops, and grasses in 2100 all showed a substantial decrease under the SSP5-8.5 scenario (Fig. 3c). Although trees still dominated the reduction of total Hg(0) uptake due to their largest global coverage (Supplementary Fig. 11), crops were the most affected by eCO₂, showing a decline of nearly 60%. Indeed, massive grass species belong to C₄ photosynthetic processes of carbon fixation in plants (C₄ plants), while trees and crops are predominantly C₃ photosynthetic processes of carbon fixation in plants (C₃ plants)[24]. The photosynthesis of the former is less limited by ambient atmospheric CO₂ concentrations and subsequently responds less to eCO₂ than the latter[25]. As C₃ plants dominate global vegetation and account for most of the Earth's current plant life[26], an overall significant weakening of global vegetation Hg(0) uptake by the biogeochemical effects of eCO₂ was found (Supplementary Fig. 12).

**Decoupled CO₂ and Hg**
We found the reduced Hg uptake predicted by the model for the future was caused by a decrease in stomatal conductance due to eCO₂. Vegetation uptakes Hg(0) via diffusion through stomatal pores, which is subsequently fixed by foliage[19,23,27]. Stomatal conductance depends on its aperture and is associated with plant physiological activities[28–30] (see Eq. 1 in "Methods"). The Medlyn model in our CLM5-Hg model can effectively simulate the response of stomatal conductance to eCO₂ (see Supplementary Figs. 7 and 8, details in the model validation section of the Supplementary Information). A 42.6% decrease in the stomatal conductance of global vegetation was projected for the year 2100 with eCO₂ under the SSP5-8.5 (business-as-usual) scenario, compared to present-day levels (Fig. 4). Our analysis identified that the sunlit stomatal conductance emerges as the key driving factor influencing the observed reduction in Hg(0) uptake. Specifically, the changes of conductance on the sunlit side of leaves were more consistent with the distribution of vegetation Hg(0) uptake than the

shaded sides, indicating the changes in the sunlit side as a primary factor contributing to the diminished global vegetation Hg(0) uptake (Fig. 1). In general, the sunlit side of leaves receives more direct sunlight and heat, prone to stomatal closure[31]. Conversely, the shaded side has a higher stomatal conductance density and is reserved for gas exchange with relatively stable aperture[32].

Our study illustrated a complex and extensive feedback mechanism between the terrestrial Hg, water and carbon cycles. The enhancement of photosynthesis is caused by the biogeochemical effect of eCO₂ is accompanied by the loss of plant water content[33]. During this process, plants adjust the stomatal aperture to reduce water transpiration and maximize water use efficiency[34,35]. The Medlyn model in our CLM5-Hg model is consistent with this optimal stomatal theory. With eCO₂, the CO₂ partial pressure at the leaf surface ($C_s$) also increased accordingly, leading to enhanced leaf photosynthesis ($A_n$) (Eq. (1)). However, $A_n$ is constrained by water potential while $C_s$ continues to increase in CLM5[29]. This resulted in reduced stomatal conductance ($g_s$) with eCO₂ (Supplementary Fig. 13), leading to decreased evapotranspiration (Fig. 4d). This process induced increased soil water storage, enhancing water use efficiency (Supplementary Fig. 14). This adaptive mechanism ultimately led to a nonlinear relationship between atmospheric CO₂ concentration and stomatal conductance. Stomatal conductance is directly related to vegetation uptake of Hg(0) in our model (Eqs. (1)–(4), for details refer to "Methods"). This relationship explained the gradual reduction in vegetation uptake of Hg(0). We observed this reduction from the pre-industrial and present-day periods to the SSP1-2.6 scenario (Supplementary Fig. 12). Yet, there was a significant decline in the Hg(0) uptake by vegetation from SSP1-2.6 to SSP3-7.0 due to the dramatic increase in atmospheric CO₂ concentration (Fig. 2). Intriguingly, the sensitivity of stomata to eCO₂ diminished gradually under the influence of long-term eCO₂ conditions. This occurred because guard cells and mesophyll tissues, which mediate stomatal movements, lead to decreases in stomatal aperture and size, culminating in physiological adaptation to higher concentrations[36,37]. Consequently, this resulted in a less pronounced decline from the SSP3-7.0 to SSP5-8.5 scenarios (Fig. 2).

A tight coupling between carbon and Hg in terrestrial ecosystems has been observed as a paradigm over the past two decades[10,38]. Conventionally, it has been postulated that rising atmospheric $CO_2$ levels would increase vegetation's photosynthesis rate, leading to the beneficial impact on plant growth, known as the $CO_2$ fertilization effect[11]. This effect is believed to enhance the concurrent absorption of both $CO_2$ and Hg(0), as suggested by Jiskra et al.[39], Obrist[40], and Schaefer et al.[41]. For example, atmospheric Hg and $CO_2$ have similar seasonal fluctuation patterns in both hemispheres, regulated by vegetation photosynthetic activity. An increase in terrestrial net primary production has been also speculated to contribute to a diminishing trend in atmospheric Hg(0) levels in the Northern Hemisphere over the past two decades[39,42,43]. Contrarily, we predicted a potential decoupling between the trends of $CO_2$ assimilation and Hg(0) uptake process by vegetation in the twenty-first century, when considering the dynamic response of vegetation physiological activities to climate change. The CLM5 model projected an increased greening of vegetation in many regions in the twenty-first century resulting from eCO2 (a.k.a. fertilization effect), evidenced by the increased leaf area index (LAI) in the northern mid-to-low latitudes and certain regions of the Southern Hemisphere (Fig. 4c). The increase in photosynthesis can simultaneously induce a state of water deficit and nutrient saturation within the plant's internal environment[44]. Therefore, under climate change, the increase in vegetation LAI may only represent an increase in leaf density or even stomatal numbers, but stomatal conductance may not necessarily increase accordingly. However, our model suggested a discernible decrease in the flux of Hg(0) uptake by vegetation in these areas (Fig. 1), reflecting the differences in $CO_2$ and Hg element during plant physiological processes especially those related to water dynamics in terrestrial ecosystems[45].

## Uncertainties

We noted significant uncertainties in our model results. First, various future climate forcing may introduce uncertainties in CLM5 simulations, such as underestimating the phenology and photosynthesis of future plants[46]. This occurred in part because the anomaly forcing method assumes that future changes (anomalies) can overlay present-day variability. Different sub-monthly variations may not accurately represent all facets of future climate changes, particularly in the presence of non-linear interactions or crossed thresholds[29,47]. Additionally, it is important to note that some data sources in our meta-analysis originate from seedling experiments. There were inherent physiological and morphological differences between young seedlings and fully mature plants, which could potentially influence the study's outcomes. However, young seedlings often exhibit more pronounced responses to environmental changes, making them suitable for detecting initial patterns and mechanisms in plant response to elevated $CO_2$. Additionally, there will be differences associated with different plant species. Thus, we suggest that future research should focus on this aspect, aiming to bridge the knowledge gap by including experiments across various growth stages and more species.

Given that vegetative stomatal absorption is a key mechanism in our model, we conducted an uncertainty analysis for the parameterization of $g_s$. We included the sensitivity analysis of five ecologically significant parameters (Medlyn_slope, slatop, leafCN, psi50, and stem_leaf) under four levels of perturbation (Supplementary Table 3). We found that the global vegetation Hg(0) uptake and $g_s$ range 1160–1370 Mg year$^{-1}$ and 29,600–41,900 μmol $H_2O$ m$^{-2}$ s$^{-1}$, respectively, with an uncertainty ratio of 17% and 21%, respectively (Supplementary Fig. 15). The coefficient of variation (CV, defined as the relative degree of change in the model output compared to the proportion of parameter changes) can reflect the magnitude of an individual parameter's contribution to uncertainty[48]. The sensitivity analysis revealed that the parameter "Medlyn slope" has the highest CV (1.01 and 1.32 for

vegetation Hg(0) uptake and $g_s$, respectively) (Supplementary Fig. 16 and Supplementary Table 4). In the Medlyn model within CLM5, the Medlyn slope, denoted as "$g_1$," plays a crucial role in controlling how stomata respond to $CO_2$ levels. It does this by determining the extent to which stomata open, based on the assimilation capacity, $CO_2$ concentration, and VPD[47]. However, the CLM5 model does not differentiate this parameter for different climate types, which induces relatively large uncertainties (6 ± 1.2%). Additionally, the stomatal conductance simulated by our model is slightly lower than the observed values (Supplementary Fig. 9), which could be caused by the uncertainty associated with this parameter. Indeed, Kauwe et al.[49] found a ~30% reduction of the annual transpiration fluxes after better constraining this parameter. This implies that the actual future decrease in Hg(0) uptake could potentially be even higher. More vegetation physiological parameters and Hg observations for different PFTs are thus needed to better constrain our model. There are also likely interaction effects among parameters.

There are still considerable uncertainties regarding the model representation of the land–atmosphere exchange of Hg at present day, which serves as a baseline for our prediction for the future. The atmospheric Hg concentrations and deposition were specified as a boundary condition, not yet dynamically modeled in a two-way coupled fashion. The feedback between land Hg emissions and their atmospheric abundance and subsequent deposition onto the land are also not considered. Although our current framework can well diagnose the direct impact of changing climate on these exchange fluxes, an online land-atmosphere coupled model will be needed to reveal a more comprehensive and accurate changes in global Hg budget in future works. Additionally, current isotopic evidence indicates that the photoreduction process is related to the re-emission of Hg(0) by vegetation leaves, with this re-emission ratio reaching nearly 30% in subtropical forest areas[50]. However, for the majority of other regions worldwide, we lack sufficient observational data to make estimates. In our model, we have only used median values as the reduction parameter[19]. Therefore, in future research, we need to utilize more measured data to refine our parameterization scheme. Our model also did not consider the absorption of Hg from underground root systems and root secretions[51]. Indeed, Hg is hard to enter the plants via the root, as most previous studies have shown[52–54]. Meanwhile, our model did not account for the translocation of Hg among plant tissue organs. A recent study suggested that a significant proportion of Hg in roots may originate from absorption by leaves and subsequent translocation, with an estimation of up to 300 Mg year$^{-1}$ of atmospheric Hg° stored in roots[55], but the specific migration and distribution mechanisms are still unclear. Furthermore, the model simplified the soil Hg processes, following GTMM (Global Terrestrial Mercury Model)[38], by focusing mainly on the microbial reduction process. It did not account for other processes like the radiative transfer in soil, photo-reduction, and other abiotic reduction processes[56,57]. These processes also have a potential influence on the amount of Hg(0) uptake by the vegetation, and could be incorporated in our model when more data is available.

In our CLM5-Hg model, throughfall primarily originates from the washing off of atmospheric divalent mercury (sum of the Hg(II) dry deposition onto the canopy surface and the Hg(II) wet deposition that has not been reduced)[38,58]. Recent studies indicated that epiphytic vegetation on canopies absorbs atmospheric Hg(0) and decomposes into humus, adhering to tree trunks and canopies, where mercury is subsequently washed into throughfall by precipitation[59]. Additionally, research indicated that the temporal scale and frequency of sampling for throughfall mercury measurements can impact the accuracy of their estimates[60]. Therefore, our model has limitations in this part, and more extensive experimental data covering broader spatiotemporal scales is needed to further constrain the model (e.g., flux measurements or isotope compositions). The anthropogenic and legacy Hg emissions from land and ocean also remained unchanged in this study.

Additional uncertainties also aroused from our current understanding of the biogeochemical response of plants to eCO$_2$ and their possible adaptability to long-term changes[24]. Overall, these uncertainties necessitate further calibration of the model when more data is available and can be effectively addressed as scientific knowledge evolves. The model should be interpreted as a diagnostic tool designed to unveil the influence of individual factors. It serves as a foundation for a more realistic and comprehensive prediction that takes into account factors such as the connection between future Hg and greenhouse gas emissions[1,61], the enhanced soil microbial activity[41], ocean warming and acidification[62], and amid many others[12,63].

## Implications

We found that, in the climate change scenario, the atmospheric Hg(0) uptake by terrestrial vegetation in 2100 will be likely to decrease by more than half compared to present-day conditions. The atmospheric CO$_2$ concentration is an important factor that will impact vegetation Hg uptake in the future. The continuous increase in CO$_2$ concentration will lead to a warming effect, which alters global precipitation patterns[64,65]. This could potentially increase VPD and cause drought in many regions, consequently affecting the Hg processes in terrestrial ecosystems. In addition, the composition of global plant communities could be modified over extended time scales[66]. For instance, persistent severe drought could lead to widespread vegetation mortality and shifts in the composition of tropical forest tree[67]. Consequently, this affects the distribution pattern and magnitude of vegetation Hg flux[68].

Our findings revealed a suppression of atmospheric Hg(0) uptake by plants across most regions in the twenty-first century due to reduced stomatal conductance in vegetation caused by increased CO$_2$. With climate change, the bypassing of atmospheric Hg(0) sequestration by plants and the deposition of foliar Hg to the soil lead to increasing concentrations in the atmosphere. This Hg can then be converted to HgII, which is deposited in aquatic ecosystems and can subsequently be methylated[6,18,54]. Furthermore, these inorganic Hg compounds are transformed into methylmercury by microbes. This process leads to the enrichment of methylmercury in riverine and marine food chains. As a result, a substantial threat to human health arises through the consumption of inland aquatic animals and seafood, including commercial fish[69–71]. These processes coincide with the changes in land use/land cover, such as the potential shift from Amazon rainforest to savannah, which also decreases the land Hg sink and contributes to an additional movement of Hg into the ocean[18], construing an additional climate change penalty via Hg cycles. Furthermore, although the impact of anthropogenic source emissions was not the focus of this study, some estimations indicated that global anthropogenic emissions of Hg will increase in the forthcoming decades under the current legislative scenario[61,72,73]. Therefore, under future climate change scenarios, it is possible that there will be a greater threat to human health. We did not consider the impacts LULCC in this study. Under global warming, vegetative succession following melting and increased precipitation intensity is likely to lead to an increase in vegetative biomass and, consequently, an increase in Hg(0) uptake by vegetation[10]. Indeed, the interactive effects of climate change combined with changes in LULCC worth further examination.

The terrestrial ecosystem, recognized as a significant Hg sink, may face disruptions under future climate change scenarios, particularly with rising atmospheric CO$_2$ concentrations[5,74]. Therefore, it becomes crucial to comprehensively consider the tight coupling among Hg, CO$_2$, and water cycles when assessing the effectiveness of the Minamata Convention within the context of climate change. From the perspective of the global Hg cycle, considering only the air-land exchange process is insufficient to achieve global mass balance. Both anthropogenic releases and the air-ocean exchange of Hg can potentially affect atmospheric Hg levels, thus influencing the air-land exchange process. Therefore, future research should aim to further

incorporate time-varying anthropogenic emissions and develop a fully coupled land-atmosphere-ocean global Hg model within CESM2. This would enable a comprehensive understanding of the complete pathway of Hg from emission to deposition.

## Methods
### CLM5-Hg

We applied a state-of-the-art global terrestrial Hg model to explore the impact of climate change on global vegetation Hg(0) uptake. We used the CLM5-Hg model, which comprehensively contains the biogeophysical and biogeochemical processes that control terrestrial Hg cycling (Supplementary Fig. 1). CLM5-Hg was tested against observational field data by Yuan et al.[19]. The model simulates the migration, transformation, accumulation, and emission processes of Hg in terrestrial ecosystems. This includes processes such as the stomatal and non-stomatal uptake of Hg(0) in leaves, the throughfall of divalent Hg [Hg(II)], and the formation of litter and soil Hg. It also includes processes such as the leaching of soil Hg, photo-reduction, as well as microbial decomposition, thermal evaporation, and emissions from wildfires.

The CLM5 model represents its surface heterogeneity with multi-layer nested grid cells. The first layer of the sub-grid consists of land units, including five types: vegetation, lakes, cities, glaciers, and crops. The second layer of the sub-grid represents soil columns, indicating the state changes of the soil within the same land unit. The third layer of the sub-grid is plant functional types (PFTs) with different biogeochemical processes. Sub-grids within the same model grid use the same atmospheric forcing dataset, but the diagnostic variables for each sub-grid are simulated independently. The vegetation covering the land surface is composed of 16 different PFTs (temperate-needleleaf evergreen tree, boreal-needleleaf evergreen tree, boreal-needleleaf deciduous tree, tropical-broadleaf evergreen tree, temperate-broadleaf evergreen tree, tropical-broadleaf deciduous tree, temperate-broadleaf deciduous tree, boreal-broadleaf deciduous tree, temperate- broadleaf evergreen shrub, temperate-Broadleaf deciduous shrub, boreal-broadleaf deciduous shrub, C$_3$ arctic grass, C$_3$ grass, C$_4$ grass, and crops)[19]. Their differences in leaves and stems determine the uniqueness of different vegetation in reflection, transmittance, and solar radiation absorption. Root distribution parameters control soil moisture absorption, aerodynamic parameters determine thermal resistance, moisture, and momentum transfer, and photosynthetic parameters determine stomatal resistance, photosynthesis, and evapotranspiration processes. All PFTs are divided into three different types of phenology: perennial evergreen types, seasonal deciduous types determined by temperature and daylight length, and multi-seasonal stress deciduous types determined by temperature and soil moisture[29].

We used the offline version of CLM5 (0.90° latitude × 1.25° longitude) with coupled biogeochemical cycles (BGC), which is forced by the dataset of Global Soil Wetness Project (GSWPS), a 3-hourly 0.5° global forcing product based on 20th Century Reanalysis version. The biogeochemical model was run for 200 years, and the CLM5-Hg was run for 10 years in each scenario simulation (1841–1850 for the pre-industrial era, 1991–2000 for the present day, and 2091-2100 for the future). The results for the last year of each simulation were used for data analysis, as this time point reached a steady state, as indicated by two representative variables (see Supplementary Fig. 17). We used the simulated atmospheric Hg(0) concentrations and the dry and wet deposition fluxes of Hg(II) from the CAM6-Chem model as its upper boundary conditions[75]. To isolate the impact of climate change, anthropogenic Hg emissions were maintained at the present-day level[76].

### Stomatal model

In CLM5-Hg, maximum stomatal conductance was obtained from the Medlyn "empirical-optimal" conductance model[30]. This stomatal model calculates stomatal conductance (g$_s$) based on net leaf

photosynthesis, the $CO_2$ concentration at the leaf surface, and the VPD. The stomatal resistance of the leaf is:

$$\frac{1}{r_s} = g_s = g_0 + 1.6\left(1 + \frac{g_1}{\sqrt{D}}\right)\frac{A_n}{C_s/P_{atm}} \qquad (1)$$

where $r_s$ is the stomatal resistance, $g_0$ is the minimum stomatal conductance, $A_n$ is leaf net photosynthesis, $C_s$ is the $CO_2$ partial pressure at the leaf surface, $P_{atm}$ is the atmospheric pressure, and $D$ is the VPD at the leaf surface. The value of $g_1$ depends on the PFTs following the CABLE model[49]. The model further have corrected $r_s$ by partitioning to sunlit and shaded side leaf stomatal resistance and the condition of snow cover[29]:

$$R_s = \left(\frac{f_{sun} \times elai}{r_s^{sun}} + \frac{(1 - f_{sun}) \times elai}{r_s^{sha}}\right)^{-1} \qquad (2)$$

where $R_s$ represents the adjusted stomatal resistance, $f_{sun}$ is the sunlit fraction of canopy, elai represents one-sided LAI buried by snow, $r_s^{sun}$ is the sunlit leaf stomatal resistance, and $r_s^{sha}$ is the shaded leaf stomatal resistance.

The dry deposition flux is used to calculate the absorption of atmospheric Hg(0) by global vegetation[77]:

$$F_d(z) = V_d C_{(z)} \qquad (3)$$

where $F_d(Z)$ represents the Hg(0) dry deposition flux at height z, $C_{(z)}$ is the atmospheric Hg(0) concentration, and $v_d$ is the dry deposition velocity calculated following the Wesely scheme[78]:

$$V_d = \frac{1}{R_a + R_b + R_c} \qquad (4)$$

where $R_a$ is the aerodynamic resistance between a specific height and the surface, $R_b$ is the quasi-laminar sublayer resistance, and $R_c$ is the bulk surface resistance. $R_c$ is mainly determined by the adjusted stomatal resistance ($R_s$).

## Experiment design

We simulated the terrestrial Hg cycling that is representative of the present-day conditions to serve as the baseline, and 1850 as the pre-industrial climate condition. We selected three different SSPs to represent three different future $CO_2$ emission scenarios in 2100: SSP1-2.6 represents the lowest, SSP5-8.5 represents the highest, and SSP3-7.0 represents a moderate scenario. To better unravel the effects of individual factors on global vegetation uptake of Hg(0), a suite of simulations was performed in the pre-industrial era and the hypothesized twenty-first century under the SSP5-8.5 scenario. To mitigate high computational costs, we adopted an alternative approach known as "anomaly forcing" for land-only simulations in CLM5 in line with future climate projections, enabling the generation of climate data to drive CLM5[79]. This method used data from a fully coupled simulation to produce monthly changes in near-surface atmospheric states and fluxes, relative to current conditions[79]. We isolated the effects of individual factors by sequentially altering only one variable to reflect future states while maintaining other variables in line with present-day conditions. To eliminate the impact of land use and land cover changes, as well as aerosols, we preprocessed the vegetation patterns and aerosol deposition in all simulation scenarios using the specified file provided by CLM5. These climate change factors specifically include atmospheric $CO_2$ concentration (limited to its biogeochemical effects and its climate effects are reflected in other factors), precipitation, radiation, meteorological factors such as temperature, humidity, pressure, and wind. In the "all" scenario, all these factors were modified simultaneously to simulate the overall effect. Additionally, we

examined the interactions between the biogeochemical effect of $CO_2$ and temperature and precipitation. We also have intentionally maintained constant levels of anthropogenic emissions and atmospheric Hg concentrations across all scenarios. This methodological choice was made to isolate and underscore the influence of climatic variables on Hg dynamics. By controlling for anthropogenic inputs, we aim to provide a clear assessment of how climate change alone can affect the biogeochemical cycling of Hg.

## Data and meta-analysis

We utilized a global vegetation Hg(0) flux dataset to constrain and validate our simulated global vegetation assimilation of Hg(0) at present day. This database contained 79 publications with measurements ranging from 1987 to 2020[18]. These data were from 37 different measurement sites across the world covering the East Asia, Western Europe, and North America regions. The modeled Hg(0) vegetation uptake flux was constrained by the 60 individual data points and agrees well with the observations ($r^2 = 0.38$)[19]. The datasets of global stomatal conductance were used to validate the accuracy of stomatal uptake process by CLM5-Hg[37].

We used meta-analysis to assess the biogeochemical effects of experimentally elevated $CO_2$ (e$CO_2$) on foliar Hg concentration. We searched for journal articles using the ISI Web of Science with the following keyword combinations: (elevated $CO_2$ concentration OR $CO_2$ enrichment OR increasing $CO_2$ concentration) AND (mercury OR mercury concentration OR mercury uptake) AND (tree OR grass OR plants OR vegetation OR leaf OR leaves) from 1990 to 2023. Papers have to meet the following criteria to be included in our dataset: (i) e$CO_2$ experiments were conducted in terrestrial ecosystems; (ii) initial environmental factors in control plots were the same as those in e$CO_2$ plots; (iii) at least two $CO_2$ concentration regimes were compared. Finally, our dataset included 368 observations from six studies[80–85]. To make sure we include all important studies, we did another search using Google Scholar and sort the studies based on their relevance.

The weighted mean response ratio (lnRR) is employed to analyze the treatment effect on vegetation mercury (Hg)[37,86], the effect size is estimated as:

$$\ln RR = \ln\left(\frac{\overline{x}_{eCO_2}}{\overline{x}_{aCO_2}}\right) \qquad (5)$$

where $\overline{x}_{eCO_2}$ and $\overline{x}_{aCO_2}$ represent the mean of the elevated $CO_2$ level and ambient $CO_2$ level, respectively.

The individual observations are assigned weights calculated from the experimental replications:

$$weight = \left(\frac{n_{eCO_2} \times n_{aCO_2}}{n_{eCO_2} + n_{aCO_2}}\right) \qquad (6)$$

where $n_{eCO_2}$ and $n_{aCO_2}$ indicate the numbers of replications at elevated $CO_2$ level and ambient $CO_2$ level, respectively.

Finally, lnRR is transformed to percentage change (%) as:

$$RR\% = (\exp(\ln RR) - 1) \times 100 \qquad (7)$$

The natural log-transformed foliar Hg concentration or flux (Hg$_{veg}$) sensitivity (lnSens) is calculated as:

$$\ln Sens = \frac{\ln RR}{\Delta} \qquad (8)$$

where lnRR represents the natural log-transformed response ration, and the $\Delta$ is the magnitude of e$CO_2$ (per 100 p.p.m. increase). The weighted means of lnSens and percentage sensitivity are calculated using equations similar to those presented in Eqs. (6) and (7) above.

A negative effect size indicate a decline in the $Hg_{veg}$ (response variable) for the treatment plots compared to the control plots. A variable is considered significantly different between the treatment and control plots ($P < 0.05$) if the 95% confidence intervals (CI) of the effect size for that variable does not overlap with zero. Differences between subgroups are deemed significant if their CIs do not overlap. Furthermore, due to the limitations of the data, it is not feasible to specifically subdivide PFTs for a direct one-to-one comparison between the model output and the observational data. Therefore, we have roughly categorized these studies into three subgroups: tree, grass, and crop (Supplementary Table 2). This categorization aims to observe the impact of $eCO_2$ on plants from different PFTs, enabling comparison with our model results.

## Uncertainty analysis

We conducted an uncertainty analysis on the Hg(0) uptake and $g_s$ using a perturbation experiment. This was based on the process of stomatal uptake, which is a crucial part in the assimilation of atmospheric Hg(0) by global vegetation[55]. Our focus was on a set of five plant physiology-related parameters within the stomatal conductance model (Medlyn model in CLM5), selected due to their mechanical impacts on responses to elevated $CO_2$ levels and their significance in representing important ecological processes in vegetation[47] (Supplementary Table 3). The Medlyn_slope is the slope of the Medlyn model (Eq. 1), where the slope parameter dictates the extent of stomatal opening based on a given mix of assimilation capacity, $CO_2$ concentration, and VPD. Slatop measures the leaf area per gram of leaf biomass. Higher SLA values, indicating thinner and more efficient leaves, lead to a greater LAI from the same biomass. Leaf_cn denotes the ideal leaf carbon to nitrogen (C:N) ratio, with higher leaf nitrogen enhancing photosynthesis but increasing respiration costs. Psi50 is the water potential at which there is a 50% loss of conductivity, a hydraulic trait of plants that effectively reflects the vegetation's state in response to water-deficient conditions. Stem_leaf determines the biomass distribution between stem and leaves. As this ratio increases, it diminishes the achievable LAI per unit of carbon and nitrogen dedicated to growth, while concurrently increasing the equilibrium woody biomass. These parameters were selected from the broader set of CLM5 parameters through a method primarily guided by the model's structure and thorough iterative testing during CLM5's development[47]. To simplify understanding and pinpointing model behavior changes due to parameter variations, we employed one-at-a-time (OAAT) perturbations from the default settings. This approach, unlike a comprehensive global parameter sensitivity analysis, enables easier visualization and interpretation of results. We assessed the sensitivity of simulations in the CLM5-Hg model under a standard scenario, testing four levels for each parameter as outlined in Supplementary Table 3, resulting in a total of 20 distinct physical perturbation ensembles. In addition, we also considered a condition in which all parameters are altered simultaneously at each level. We estimated the relative uncertainty in Hg(0) uptake and stomatal conductance using the CVs, defined as the relative degree of change in the model output compared to the proportion of parameter changes[48].

## Model validation

We curated a dataset within the $CO_2$ concentration ranges projected by our model under three SSP scenarios, which included 462 paired observations of treated versus control groups for various species across different vegetation biomes (Supplementary Figs. 6 and 7). First, the comparison results under the current scenario indicated that the simulation can captures the global pattern of stomatal conductance ($r > 0.5$, Supplementary Fig. 8). Second, we performed point-by-point validations against matched observational data for projected $eCO_2$ concentrations in each scenario, achieving an $r$ value of 0.50 (Supplementary Fig. 9).

Our previous research has shown that the CLM5-Hg model can capture the global distribution of vegetative Hg and litter Hg concentrations well ($r > 0.6$) via the Hg(0) dataset and a large number of observational datasets related to vegetative Hg tissue concentration[19].

### Reporting summary

Further information on research design is available in the Nature Portfolio Reporting Summary linked to this article.

## Data availability

All data generated or analyzed are available in the main text, the Supplementary information, and the research group website: https://www.ebmg.online/mercury. GSWP3 climate dataset for CESM2: https://svn-ccsm-inputdata.cgd.ucar.edu/trunk/inputdata/atm/datm7/. The data from the uncertainty analysis and the sensitivity experiment of model generated in this study are provided in the Supplementary Information/Source data file. Source data are provided with this paper.

## Code availability

All core CLM5-Hg model code is available at the research group website: https://www.ebmg.online/mercury. The CESM2 code: https://github.com/ESCOMP/CESM. The CLM5 code: https://github.com/ESCOMP/CTSM/tree/master/src.

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

## Acknowledgements

We thank Guiyao Zhou, Wenbin Chen, Huimin Zhou, Xizhen Xia, Chao Zhang, and Xiaojiang Liu for the helpful discussions and suggestions. This work was supported by the the National Natural Science Foundation of China (NSFC) 42394094, the "GeoX" Interdisciplinary Research Funds for the Frontiers Science Center for Critical Earth Material Cycling, Nanjing University, the Fundamental Research Funds for the Central Universities (grant nos. 14380188, 14380168), the Frontiers Science Center for Critical Earth Material Cycling, and the Collaborative Innovation Center of Climate Change, Jiangsu Province.

## Author contributions

Conceptualization: T.Y., Y.Z., S.H. Methodology: T.Y., Peng Zhang, Y.Z., J.G., P.W., Yujuan Wang, W.G. Investigation: T.Y., J.S., Peipei Zhang, Z.S., X.M., D.P., Q.P., Y.Z., Yabo Wang. Visualization: T.Y., Peng Zhang, Yujuan Wang, Y.Z., S.H. Funding acquisition: Y.Z. Project administration: Y.Z. Supervision: Y.Z. Writing—original draft: T.Y, Y.Z. Writing—review and editing: T.Y., Y.Z, J.G., Z.S., X.M., J.S., Peng Zhang, H.G., W.G.

## Competing interests

The authors declare no competing interests.
