## [Peer Review File · Nature Communications]

Potential decoupling of CO₂ and Hg uptake process by global vegetation in the 21st centuryREVIEWER COMMENTS

Reviewer #1 (Remarks to the Author):

Review for “Decoupling of CO₂ and Hg uptake by global vegetation in the 21st century” by Yuan et. al. (NCOMMS-24-07636-T).

The manuscript describes the modeling findings on the change of carbon dioxide (CO₂) and gaseous elemental mercury (Hg₀) uptake by plants under several climate scenarios using the NCAR Community Land Model 5 with Hg processors (CLM5-Hg). The primary conclusion from the model results is that Hg uptake by vegetation is expected to decrease significantly through 2100 under a business-as-usual climate scenario because of decreasing stomatal conductance with increasing concentration of CO₂; and that global Hg uptake by vegetation, which has been thought to be closely associated with the global carbon cycle, will be gradually “decoupled” from CO₂ assimilation caused by vegetation.

This is an interesting study that attempts to describe the long-term trend of an important sink term of atmospheric Hg(0) facilitated by foliar uptake in the terrestrial ecosystem. The work is relevant considering the concerted efforts devoted to source elimination of atmospheric Hg release regulated by the Minamata Convention of Mercury. However, it is not completely clear what scientific processors are incorporated in the CLM5-Hg (not even in Yuan et. al., 2023, <https://doi.org/10.1016/j.envint.2023.107904>, where the model was described), making the results of such long-term simulation doubtful and somewhat difficult to evaluate. Another technical concern is that the stated “Hg(0) uptake by vegetation” is not clearly defined and the presented “present-day uptake (3,138 Mg/yr)” appears to be much greater than what is currently understood. It is also unclear if the release and re-emission from the bare soil surface are included in the calculation. Furthermore, since the biogeochemical cycle of Hg in vegetated ecosystems is still relatively poorly understood, biases in model results can propagate over a long simulation period, which is the case in this work. These concerns, and the associated uncertainties of the modeling framework, do not project confidence that the conclusion would be valid. Finally, there is a general lack of testable hypotheses in this work, and the study objectives are not laid out specifically.

Overall, I recommend a major revision of the manuscript before consideration for publication. Specific comments are as follows:

1. In the introduction, provide the study hypotheses and the associated study objectives to justify the need for this work.
2. Clearly describe what processors are included in the CLM5-Hg model. What is equally important is what fluxes are being simulated. Is it only Hg(0) uptake by vegetation, or the underlying soil release and re-emission are also included? What are the vegetation types? What soil chemistry and surface redox chemistry are considered in the model? Are wet deposition terms of Hg(II) included in the model? How are throughfall fluxes calculated? Is the immediate reflux of deposited Hg considered? Substantial modeling details are missing in this work and Yuan et. al. (2023). Without understanding the

parameterizations, it is difficult to evaluate and verify the model results.

3. In Figure 1a, what are the “uptake” terms included in the graph? It appears that there is substantial “vegetation uptake” in regions where there is no vegetation. The reported value is much greater than what is currently known (e.g., about 1200 Mg/yr) based on litterfall data. If throughfall is included, how is it estimated given the general lack of through measurement for model verification?

4. It is unclear how the elevated CO₂ would lead to changes in the surface resistance terms (particularly the stomatal resistance) and how it would lead to the decoupling claimed in the manuscript. A mechanistic explanation will benefit understanding of the underlying processes forced by the changed climate.

5. The trend shown in Figure 2, in addition to the uptake quantity in question, does not appear to be consistent with the current understanding of the future trend of vegetation Hg(0) uptake under a warming climate. A warming climate will lead to melting of surface ice and permafrost soil. The vegetative succession after the melting and the increased intensity of precipitation will lead to a substantial increase of vegetative biomass that increases the Hg(0) uptake by vegetation. The model results are counter-intuitive and there is little plausible explanation provided in the manuscript.

6. Similarly in Figure 3, it is possible the land-cover changes under the changed climate would lead to significantly different vegetation types and distribution. Is the effect of land-use changes considered in the modeling?

7. In the implication section, it is not clear how the “complex and extensive feedback mechanisms between the terrestrial Hg, water and C cycles” are illustrated in this work. The discussion can benefit from a more specific discussion on what processes are responsible for the simulated changes of the model results, and why the simulated results and the temporal trends appear to be significantly different from the assessment in earlier studies.

Reviewer #2 (Remarks to the Author):

I liked that the authors of this modeling paper were very honest about the limitations of their model calling this effort a diagnostic exercise. They have provided a nice summary of how changes in CO₂ concentrations and climate change could impact Hg uptake. I feel they were very honest about the limitations of this study.

Some limitations not addressed clearly include impacts of deforestation that is only mentioned briefly in the implications section.

They note that plants physiologically could adapt to higher concentrations. Is there any work to indicate how plant behavior could change?

They are clear that the experimental work was done with seedlings, and there will be differences associated with different plant species.

Line 287 it has been demonstrated in multiple studies that Hg does not enter the plants via the roots and that roots act as a barrier accumulating soil Hg

Line 320 I would change this to read “bypass sequestration of Hg₀ by plants and deposition of foliar Hg to the soil causing increasing concentrations in the atmosphere, where it could be converted to Hg_{II} that

is deposited to ecosystems, where it can subsequently be methylated.

Gaseous elemental Hg is the form taken up by foliage. HgII is not taken up see Arnold et al. (2018)

Some general housekeeping

Line 28 which should be that.

Line 48 previous research has

Line 84 and throughout the text what was done should be described in past tense. Here indicated should be used. Line 89 were instead of are.

References for Supplemental Information are not in alphabetical order and

Arnold J, Gustin MS, Weisberg PJ. Evidence for Nonstomatal Uptake of Hg by Aspen and Translocation of Hg from Foliage to Tree Rings in Austrian Pine. *Environmental Science & Technology* 2018; 52: 1174-1182.

Reviewer #3 (Remarks to the Author):

Review of the manuscript NCOMMS-24-07636-T titled "Decoupling of CO₂ and Hg Uptake by Global Vegetation in the 21st Century."

The manuscript addresses a crucial topic, given the pivotal role of plants in the global cycle of toxic mercury and the current and anticipated changes in CO₂ emissions and budget. Its significance is heightened by the imperative to effectively implement the UN Minamata Convention on Mercury. Despite numerous studies on the intricate impact pathways of climate change on vegetation Hg(0) uptake, the cascading effects of climate change-related factors on the terrestrial Hg cycle, particularly the uptake of Hg(0) by vegetation, remain unclear.

The study employs a compelling combination of the Community Land Model - Hg (CLM5-Hg) and the Community Earth System Model (CESM). Although the novel method has been previously applied by the authors (Yuan et al., 2023), the current manuscript utilizes a distinct approach, yielding entirely new results. The presented findings are highly valuable and align with the global trend of research on pollutant cycling and its connections to climate change. The publication is meticulously and thoughtfully prepared, and its content is certain to capture readers' interest.

The introduction section is succinct yet effectively articulates the rationale for the research topic. The results and discussion section is well-organized, comprising four subsections that sequentially address changes in Hg uptake by vegetation, the coupling of terrestrial Hg and CO₂ uptake, model uncertainties, and potential implications. In this section, my comments primarily relate to the part associated with model uncertainty. However, they do not question the substantive value of the paper. My comments are driven more by scientific curiosity in this context. A separate section constitutes the Materials and Methods part, in which the authors provide a detailed description of the CLM5-Hg model. This model simulates the migration, transformation, accumulation, and emission processes of Hg in terrestrial

ecosystems. It encompasses both stomatal and non-stomatal uptake in leaves, throughfall, soil formation and leaching, photo-reduction, microbial decomposition, thermal evaporation, and emissions from wildfires. In my assessment, the model's components allow for a comprehensive tracking of the Hg cycle involving land plants. Particularly interesting is the application of distinguishing numerous plant functional types (PFTs) characterized by different leaf, stem, and root properties. This approach has been successfully tested against observational field data in a previous study by the authors (Yuan et al., 2023). The data used regarding simulated atmospheric Hg(0) concentrations, as well as the dry and wet deposition fluxes of Hg(II), do not raise concerns, according to Zhang & Zhang (2022). Similarly, the utilization of the global vegetation Hg(0) flux dataset (Feinberg et al., 2022; Yuan et al., 2023), as well as CO₂ dataset (Millhollen et al., 2006; Sañudo-Wilhelmy et al., 2008; Stamenkovic and Gustin, 2009; Duval et al., 2011; Demers et al., 2013; Tang et al., 2021), is adequate. The uncertainty analysis appears to have been conducted thoroughly and comprehensively in my opinion.

In conclusion, the manuscript is presented in a clear and generally grammatically correct manner. However, I advise authors to carefully check through the text to ensure consistency in verb tenses, especially when discussing model results or future research directions.

Below, you will find a list of my suggestions.

Introduction

Lines 35-36: Consider adding information on the adverse effects of Hg on the ecosystem

Lines 39-42: Consider adding information that long-living vegetation can store not only present-day Hg but also Hg emitted into the atmosphere decades ago

Line 42: Change "warming temperatures" to "rising temperatures". The same applies to line 51

Line 45: Add a comma after "vegetation"

Lines 51-52: The symbol "Hg" has already been explained at the beginning of the Introduction

Line 60: Correct the citation for "Zhang et al., 2016"

Line 74: What is the source of pre-industrial data? Also, correct "c.a" to "ca."

Line 82: Change to "Materials and Methods"

Results and Discussion

Line 86: Consider specifying what kind of scenarios you are referring to, in this case "business-as-usual".

The same applies to line 188

Lines 106-107: Consider adding "2°C scenario" and "regional rivalry". The same applies to line 213

Line 138: Provide a clear understanding of what the percentages represent, e.g., a change in Hg uptake or a decrease in foliage Hg levels

Line 179: Change to "Decoupled CO₂ and Hg" since you are referring to CO₂ specifically rather than total carbon. The same applies to lines 232, 305, and 333

Lines 189-192: Consider providing a concise statement that summarizes the key finding before delving into the specific details. This can help guide the reader's understanding

Lines 204-207: Consider breaking down this sentence into shorter ones for better clarity. The same applies to lines 208-212

Lines 217-220: Consider providing a brief transition sentence. Also, please explain "CO₂ fertilization

effect”

Lines 225-227: Please clarify whether you mean a decoupling in trends or a decoupling in the magnitude of the trends

Lines 247-249: Please rewrite this sentence for better clarity, e.g., "This occurs in part because the anomaly forcing method assumes that future changes (anomalies) can overlay present-day variability."

Lines 252-253: Consider providing a brief transition sentence, e.g., "Additionally, it's important to note that some data sources in our meta-analysis originate from seedling experiments."

Lines 271-274: Please rewrite this sentence for better clarity. Also, please be consistent in terminology used, either "Medlyn" or "medlyn"

Line 283-285: The statement lacks specificity. Please provide more details on the specific uncertainties in the model representation of land-atmosphere Hg exchange and its implications for future predictions

Lines 285-287: Please clarify the significance of these processes in the context of your study. How do they contribute to the uncertainties mentioned earlier?

Lines 287-288: Please explain the rationale behind excluding the absorption of Hg from underground root systems and root secretions. This omission could have implications for the overall accuracy of your model

Line 288-289: Consider elaborating on why anthropogenic and legacy Hg emissions remain unchanged. Provide justification or discuss potential impacts on the study's outcomes.

Lines 317-319: Consider rephrasing the beginning for smoother flow, e.g., "Our findings reveal a suppression of atmospheric Hg(0) uptake by plants across most regions in the 21st century due to reduced stomatal conductance in vegetation caused by increased CO₂."

Lines 321-325: Consider breaking down this sentence into shorter ones for better clarity

Lines 330-335: Consider rephrasing to enhance clarity, e.g., "The terrestrial ecosystem, recognized as a significant Hg sink, may face disruptions under future climate change scenarios, particularly with rising atmospheric CO₂ concentrations. It becomes crucial to comprehensively consider the tight coupling among Hg, CO₂, and water cycles when assessing the effectiveness of the Minamata Convention within the context of climate change."

Figures

Figure 1: Please add in the caption the unit of time (yr⁻¹) for the global total values of Hg(0) uptake. In line 101 change to "Materials and Methods"

Figure 2: Please add explanations to the scenario numbers in the legend.

Figure 3: What does the purple triangle on Figure 3a mean?

Figure 4: It seems there is an error in the color scale legend of figures 4a and 4c. Now it is 0 - 3 - 2 - 1 instead of 0 - 1 - 2 - 3

Materials and Methods

Lines 613-619: From what does the division of the soil pool into specific layers in the model result? Has this approach been applied before, and if so, could you please provide references?

Lines 630-631: One of the model assumptions is that anthropogenic mercury (Hg) will persist at the current level (Zhang et al., 2016). However, some estimations indicate that global anthropogenic emissions of Hg will increase in the forthcoming decades under the current legislative scenario, e.g. Rafaj

et al. (2013, <http://dx.doi.org/10.1016/j.atmosenv.2013.06.042>), Pacyna et al. (2016, <https://doi.org/10.5194/acp-16-12495-2016>), Brocza et al. (2024, <https://doi.org/10.5194/egusphere-2024-41>). Even if this rise won't be substantial compared to the present-day level, I believe it would be worthwhile to consider this aspect somewhere in the manuscript.

Reviewer #1 (Remarks to the Author):

The manuscript describes the modeling findings on the change of carbon dioxide (CO₂) and gaseous elemental mercury (Hg⁰) uptake by plants under several climate scenarios using the NCAR Community Land Model 5 with Hg processors (CLM5-Hg). The primary conclusion from the model results is that Hg uptake by vegetation is expected to decrease significantly through 2100 under a business-as-usual climate scenario because of decreasing stomatal conductance with increasing concentration of CO₂; and that global Hg uptake by vegetation, which has been thought to be closely associated with the global carbon cycle, will be gradually “decoupled” from CO₂ assimilation caused by vegetation.

This is an interesting study that attempts to describe the long-term trend of an important sink term of atmospheric Hg(0) facilitated by foliar uptake in the terrestrial ecosystem. The work is relevant considering the concerted efforts devoted to source elimination of atmospheric Hg release regulated by the Minamata Convention of Mercury. However, it is not completely clear what scientific processors are incorporated in the CLM5-Hg (not even in Yuan et. al., 2023, <https://doi.org/10.1016/j.envint.2023.107904>, where the model was described), making the results of such long-term simulation doubtful and somewhat difficult to evaluate. Another technical concern is that the stated “Hg(0) uptake by vegetation” is not clearly defined and the presented “preset-day uptake (3,138 Mg/yr)” appears to be much greater than what is currently understood. It is also unclear if the release and re-emission from the bare soil surface are included in the calculation. Furthermore, since the biogeochemical cycle of Hg in vegetated ecosystems is still relatively poorly understood, biases in model results can propagate over a long simulation period, which is the case in this work. These concerns, and the associated uncertainties of the modeling framework, do not project confidence that the conclusion would be valid. Finally, there is a general lack of testable hypotheses in this work, and the study objectives are not laid out specifically.

Overall, I recommend a major revision of the manuscript before consideration for publication.

Response: Thank you very much for your recognition and detailed and constructive feedback on our manuscript. We appreciate the time and effort you have devoted to reviewing our work. Your comments have provided us with valuable insights that will undoubtedly improve the quality and clarity of our research. We have carefully considered each point you raised and have made corresponding revisions to address these concerns in the revised manuscript. Additionally, we have included diagrams and equations in the supplementary materials, as detailed in our responses to your specific comments below:

Specific comments are as follows:

1. In the introduction, provide the study hypotheses and the associated study objectives to justify the need for this work.

Response: We thank you for pointing out this issue. We have added the following sentences to the beginning of the last paragraph in the introduction.

Lines 63-66 “Therefore, our research aims to examine how vegetation-regulated atmospheric Hg(0) deposition will change under the impact of future climate change. We hypothesize that future climate change will increase the global atmospheric Hg(0) deposition, as suggested by previous studies (Wang 2020a; Wu et al. 2012).”

Additionally, we have added the following conclusive sentence at the beginning of the implications section.

Lines 357-359: “We found that, in the climate change scenario, the atmospheric Hg(0) uptake by terrestrial vegetation in 2100 will be likely to decrease by more than half compared to present-day conditions.”

2. Clearly describe what processors are included in the CLM5-Hg model. What is equally important is what fluxes are being simulated. Is it only Hg(0) uptake by vegetation, or the underlying soil release and re-emission are also included? What are the vegetation types? What soil chemistry and surface redox chemistry are considered in the model? Are wet deposition terms of Hg(II) included in the model? How are throughfall fluxes calculated? Is the immediate reflux of deposited Hg considered? Substantial modeling details are missing in this work and Yuan et. al. (2023). Without understanding the parameterizations, it is difficult to evaluate and verify the model results.

Response: Thank you very much for this great suggestion, which we believe helps the readers to better understand our work. Our currently developed CLM5-Hg is a mechanistic model based on a relatively complete understanding of the Hg processes in terrestrial ecosystems, with the underlying soil release and re-emission already included (see Supplementary Fig.1, as added below). However, our current investigation is primarily concentrated on vegetative uptake of Hg(0), since vegetation within terrestrial ecosystems absorbs gaseous elementary Hg [Hg(0)], serving as the largest removal mechanism of atmospheric Hg and the pivotal process within the terrestrial Hg cycle, given that vegetation typically responds more sensitively to climate change than soil does. Regarding the description of the mechanistic processes of CLM5-Hg that you mentioned, I will provide a detailed explanation for each point as follows:

i. Specifically, the CLM model includes simulations for more than 15 types of Plant Functional Types (PFTs). However, due to the limitations of observational data under our eCO₂ control experiment conditions, it is not feasible to subdivide PFTs for a direct one-to-one comparison. Therefore, in the validation process, we can only summarize the meta-sourced observational data according to their PFTs roughly and categorize them into three major groups: tree, grass, crop. Thus, referring to them as "vegetation type" was indeed inappropriate, and we have reverted to using "PFTs". The error has been corrected, and detailed descriptions have been added to the methods section to refine the corresponding part of the original manuscript:

Lines 537-542: “Furthermore, due to the limitations of the data, it is not feasible to specifically subdivide PFTs for a direct one-to-one comparison between the model output and the observational data. Therefore, we have roughly categorized these studies into three subgroups: tree, grass, and crop (Table S2). This categorization aims to observe the impact of eCO₂ on plants from different PFTs, enabling comparison with our model results.”

Line 152: “vegetation type” has been revised to “PFTs”.

ii. We largely followed the Global Terrestrial Mercury Model (GTMM) by Smith-Downey et al. (2010) to simulate the soil chemistry and surface redox chemistry, with a few modifications. Compared with previous models, our model includes a vertically resolved soil biogeochemistry scheme. This scheme features base decomposition rates that vary with depth and are modified by soil temperature, water, and oxygen limitations. It also includes vertical mixing of soil carbon and

nitrogen due to bioturbation, cryoturbation, and diffusion. The above supplementary content has been added to the revised Supplementary Information (SI) as follows:

Lines 48-75 (SI): “The decomposition process of soil Hg is tied to the soil carbon pool, assuming that the Hg in the soil binds with soil carbon pools of different ages (Smith-Downey et al., 2010). The transformation between different Hg pools and the microbial transformation rate are characterized using the conversion and respiration rates of the soil carbon pool in every layer (Lawrence et al., 2019). Upon each transfer of carbon and Hg between pools, a fraction is lost—carbon as CO₂ respiration and Hg as Hg(0) evasion into the atmosphere. Following microbial decay, we estimate that 16% of atmospheric Hg(0) evades, with the remainder being reincorporated into organic matter (Schaefer et al., 2020; Smith-Downey et al., 2010). In the framework of the single-level model structure, the foundational equation within decomposing Hg pools is as follows:

(1)

Where C_i is the Hg content of pool i , R_i represent the Hg inputs from plant tissues directly to pool i (only non-zero for litter pools and CWD), k_i is the decay constant of carbon pool i ; T_{ji} indicates the fraction of carbon directed from pool j to pool i , with a fraction r_j being lost as a respiration flux along the way. f_{Hg} is the constant representing the evasion into the atmosphere as Hg(0).

Incorporating the vertical dimension into the decomposition dynamics alters the balance equation as detailed below:

(2)

Where $C_i(z)$ is now defined at each model level in volumetric terms ($gC\ m^{-3}$), along with $R_i(z)$ and $k_j(z)$. Additionally, vertical transport is accounted for by the last two terms, representing diffusive and advective transport, respectively. In the base model, advective transport is set to zero, leaving only a diffusive flux, with diffusivity $D(z)$ defined for all decomposing carbon and Hg pools.

Our model includes a vertically resolved soil biogeochemistry scheme in CLM5 was introduced. This scheme features base decomposition rates that vary with depth and are modified by soil temperature, water, and oxygen limitations. It also includes vertical mixing of soil carbon and nitrogen due to bioturbation, cryoturbation, and diffusion (Lawrence et al., 2019).”

iii. Wet deposition of Hg(II) are also include in our model following the GTMM, and the monthly flux data are speciated from the CAM6-Chem-Hg (Zhang and Zhang, 2022). Additionally, we do not consider the immediate reflux of deposited mercury; instead, we focus on the gross uptake of Hg(0), although our model includes the re-emission process. The related supplementary content has been added to the revised SI and manuscript as follows:

Lines 36-40 (SI): “The wet deposition process of Hg(II) is simulated following the GTMM (Smith-Downey et al., 2010). The Hg(II) wet deposition fluxes are speciated from the CAM6-Chem-Hg (Zhang and Zhang, 2022). After being removed from the surfaces of leaves and soil, Hg(II) can penetrate the soil and has the ability to attach to reduced sulfur groups present in organic matter. Throughfall flux are calculated as the sum of the Hg(II) dry deposition onto the canopy surface and the Hg(II) wet deposition that has not been reduced (Smith-Downey et al., 2010).”

Lines 91-93: “This uptake represents the gross uptake of atmospheric Hg(0) through both stomatal

and cuticular (non-stomatal) processes and does not include the immediate re-emission from foliage.”

Lines 108-110: “the value represents the gross uptake of atmospheric Hg(0) through both stomatal and cuticular (non-stomatal) processes and does not include the immediate re-emission from foliage.”

Lines 30-35 (SI): “The nonstomatal uptake pathway entails the direct absorption of atmospheric Hg vapor via the cuticles in the epidermis of the canopy's upper layers (Wesely, 1989). We assume a 10% re-emission of Hg from foliage due to the release and subsequent reduction of previously sequestered Hg(0) within leaf tissue, along with a 15% re-emission from Hg deposited on leaf surfaces, transformed to Hg(0) through photoreduction (Demers et al., 2013; Yu et al., 2016; Yuan et al., 2019).”

Lines 337-346: “In our CLM5-Hg model, throughfall primarily originates from the washing off of atmospheric divalent mercury (sum of the Hg(II) dry deposition onto the canopy surface and the Hg(II) wet deposition that has not been reduced) (Paige Wright et al., 2016; Smith-Downey et al., 2010). Recent studies indicate that epiphytic vegetation on canopies absorbs atmospheric Hg(0) and decomposes into humus, adhering to tree trunks and canopies, where mercury is subsequently washed into throughfall by precipitation (Wang et al., 2020b). Additionally, research indicates that the temporal scale and frequency of sampling for throughfall mercury measurements can impact the accuracy of their estimates (Choi et al., 2008). Therefore, our model has limitations in this part, and more extensive experimental data covering broader spatiotemporal scales is needed to further constrain the model(e.g., flux measurements or isotope compositions).”

Supplementary Fig.1 The terrestrial Hg cycle as simulated by CLM5-Hg, modified from Yuan et al. (2023).

3. In Figure 1a, what are the “uptake” terms included in the graph? It appears that there is substantial “vegetation uptake” in regions where there is no vegetation. The reported value is much greater than what is currently known (e.g., about 1200 Mg/yr) based on litterfall data. If throughfall is included, how is it estimated given the general lack of through measurement for model verification? Response: Thank you for pointing these out. "uptake" refers to the dry deposition of Hg(0) driven by global vegetation. It represents the gross uptake of atmospheric Hg(0) through both stomatal and

cuticular (non-stomatal) processes, and does not account for immediate reflux of foliage. Our simulation results agree well with the global vegetation distribution as follows: areas represented with values in the figure indeed correspond to actual vegetation presence (Southworth et al., 2023).

I wonder if you might be questioning the presence of vegetation in tundra regions? Previous studies have shown that vegetation in tundra areas, such as mosses, can absorb atmospheric Hg(0) (Obrist et al., 2017; Wang et al., 2020a). The 1200 Mg/yr value based on litterfall data is very likely underestimated since these data primarily represent foliar Hg over the period near the growing season and do not account for the amount from the tissues, mosses, and lichens (Feinberg et al., 2022). Therefore, it cannot accurately estimate reversely to the Hg(0) vegetation uptake. This point was mentioned in our previous study (Yuan et al., 2023).

In our CLM5-Hg model, throughfall flux are calculated as the sum of the Hg(II) dry deposition onto the canopy surface and the Hg(II) wet deposition that has not been reduced. It is partly refer to our response to previous comments.

The above-mentioned content has been added to the revised manuscript as follows:

Lines 91-93: “This uptake represents the gross uptake of atmospheric Hg(0) through both stomatal and cuticular (non-stomatal) processes and does not include the immediate re-emission from foliage.”

Lines 108-110: “a, Hg(0) vegetation uptake flux at present-day, the value represents the gross uptake of atmospheric Hg(0) through both stomatal and cuticular (non-stomatal) processes and does not include the immediate re-emission from foliage.”

Lines 337-346: “In our CLM5-Hg model, throughfall primarily originates from the washing off of atmospheric divalent mercury (sum of the Hg(II) dry deposition onto the canopy surface and the Hg(II) wet deposition that has not been reduced) (Paige Wright et al., 2016; Smith-Downey et al., 2010). Recent studies indicate that epiphytic vegetation on canopies absorbs atmospheric Hg(0) and decomposes into humus, adhering to tree trunks and canopies, where mercury is subsequently washed into throughfall by precipitation (Wang et al., 2020b). Additionally, research indicates that the temporal scale and frequency of sampling for throughfall mercury measurements can impact the accuracy of their estimates (Choi et al., 2008). Therefore, our model has limitations in this part, and more extensive experimental data covering broader spatiotemporal scales is needed to further constrain the model(e.g., flux measurements or isotope compositions)”

4. It is unclear how the elevated CO₂ would lead to changes in the surface resistance terms (particularly the stomatal resistance) and how it would lead to the decoupling claimed in the manuscript. A mechanistic explanation will benefit understanding of the underlying processes forced by the changed climate.

Response: Regarding the process by which **elevated CO₂ leads to changes** in leaf stomatal resistance (stomatal conductance), we elaborated in detail in the section 'Decoupled CO₂ and Hg': Lines 219-227: “The Medlyn model in our CLM5-Hg model is consistent with this optimal stomatal theory. With eCO₂, the CO₂ partial pressure at the leaf surface (C_s) also increases accordingly, leading to enhanced leaf photosynthesis (A_n) (equation (1)). However, A_n is constrained by water potential while C_s continues to increase in CLM5 (Lawrence et al., 2019). This resulted in reduced stomatal conductance (g_s) with eCO₂ (Supplementary Fig. 13)...This adaptive mechanism ultimately leads to a nonlinear relationship between atmospheric CO₂ concentration and stomatal conductance.”

As for the **decoupling claimed** in our study, we elaborated it at the end of the section. We also added descriptions related to the trade-offs between the effects by increasing LAI and reducing g_s (stomatal conductance) to the original text. The integrated explanation is as follows:

Lines 250-263: Contrarily, we predicted a decoupling between the trends of CO₂ assimilation and Hg(0) uptake by vegetation in the 21st century. The CLM5 model projects an increased greening of vegetation in many regions in the 21st century resulting from eCO₂ (a.k.a. fertilization effect), evidenced by the increased leaf area index (LAI) in the northern mid-to-low latitudes and certain regions of the Southern Hemisphere (Fig. 4c). The increase in photosynthesis can simultaneously induce a state of water deficit and nutrient saturation within the plant's internal environment (Victoria et al., 2010). Therefore, under climate change, the increase in vegetation LAI may only represent an increase in leaf density or even stomatal numbers, but stomatal conductance may not necessarily increase accordingly. Our model suggests a discernible decrease in the flux of Hg(0) uptake by vegetation in these areas (Fig. 1), reflecting the differences in CO₂ and Hg elements during plant physiological processes especially those related to water dynamics in terrestrial ecosystems (Wang et al., 2021).

5. The trend shown in Figure 2, in addition to the uptake quantity in question, does not appear to be consistent with the current understanding of the future trend of vegetation Hg(0) uptake under a warming climate. A warming climate will lead to melting of surface ice and permafrost soil. The vegetative succession after the melting and the increased intensity of precipitation will lead to a substantial increase of vegetative biomass that increases the Hg(0) uptake by vegetation. The model results are counter-intuitive and there is little plausible explanation provided in the manuscript.

Response: We appreciate the reviewer for pointing out the issues related to a warming climate and for providing many excellent and constructive suggestions. We completely agree that it is very likely that vegetative succession following melting and increased precipitation intensity will lead to an increase in vegetative biomass and, consequently, an increase in Hg(0) uptake by vegetation. Our study did not consider the impacts of Land Use and Land Cover Change (LULCC), because we focused more on the direct effects of climate change (e.g. eCO₂) in this research. This point is very

insightful, and our next work is addressing this aspect. We clearly stated this point in the revised manuscript as follows:

Lines 86-87: "...and remove the effects of land use and land cover change (LULCC) and aerosols (see Materials and Methods)."

Lines 385-390: "We did not consider the impacts LULCC in this study. Under global warming, vegetative succession following melting and increased precipitation intensity is likely to lead to an increase in vegetative biomass and, consequently, an increase in Hg(0) uptake by vegetation (Wang et al., 2020a). Indeed, the interactive effects of climate change combined with changes in LULCC worth further examination."

6. Similarly in Figure 3, it is possible the land-cover changes under the changed climate would lead to significantly different vegetation types and distribution. Is the effect of landuse changes considered in the modeling?

Response: The results presented in Figure 3 are also based on OAAT analysis focusing solely on the influence of CO₂. In this study, we didn't considered the effects of LULCC, please refer to our response to last comment and

Lines 480-482: "To eliminate the impact of land use and land cover changes, as well as aerosols, we preprocessed the vegetation patterns and aerosol deposition in all simulation scenarios using the specified file provided by CLM5." Your suggestion is very valuable, and in future research, we could explore the interactive effects of LULCC and climate change, which could be an intriguing study."

7. In the implication section, it is not clear how the "complex and extensive feedback mechanisms between the terrestrial Hg, water and C cycles" are illustrated in this work. The discussion can benefit from a more specific discussion on what processes are responsible for the simulated changes of the model results, and why the simulated results and the temporal trends appear to be significantly different from the assessment in earlier studies.

Response: Thank you for pointing this out. The sentence was originally intended to summarize the results and discussion sections from before, and placing it here was indeed somewhat inappropriate. Therefore, we have moved it to Line 215 of the "Decoupled CO₂ and Hg" section in the revised manuscript. In this section, we thoroughly discuss what processes are responsible for the simulated changes in the model results and why our findings differ significantly as follows:

Lines 215-263: "Our study illustrates a complex and extensive feedback mechanism between the terrestrial Hg, water and carbon cycles. The enhancement of photosynthesis caused by the biogeochemical effect of eCO₂ is accompanied by the loss of plant water content (Katul et al., 2012). During this process, plants adjust the stomatal aperture to reduce water transpiration and maximize water use efficiency (Gardner et al., 2023; Hsiao et al., 2019). The Medlyn model in our CLM5-Hg model is consistent with this optimal stomatal theory. With eCO₂, the CO₂ partial pressure at the leaf surface (C_s) also increases accordingly, leading to enhanced leaf photosynthesis (A_n) (equation (1)). However, A_n is constrained by water potential while C_s continues to increase in CLM5 (Lawrence et al., 2019). This results in reduced stomatal conductance (g_s) with eCO₂ (Supplementary Fig. 13), leading to decreased transpiration (Fig. 4d). This process induces increased soil water storage, enhancing water use efficiency (Supplementary Fig. 14). ...A tight coupling between carbon and Hg in terrestrial ecosystems has been observed as a paradigm over the

past two decades (Smith-Downey et al., 2010; Wang et al., 2020a). Conventionally, it has been postulated that rising atmospheric CO₂ levels would increase vegetation's photosynthesis rate, leading to the beneficial impact on plant growth, known as the CO₂ fertilization effect (Liu et al., 2019). This effect is believed to enhance the concurrent absorption of both CO₂ and Hg(0), as suggested by Jiskra et al. (2018), Lawrence et al. (2019), Obrist (2007), and Schaefer et al. (2020). ...Contrarily, we predict a decoupling between the trends of CO₂ assimilation and Hg(0) uptake by vegetation in the 21st century, **when considering the dynamic response of vegetation physiological activities to climate change**. ...The increase in photosynthesis can simultaneously induce a state of water deficit and nutrient saturation within the plant's internal environment (Victoria et al., 2010). Therefore, under climate change, the increase in vegetation LAI may only represent an increase in leaf density or even stomatal numbers, but stomatal conductance may not necessarily increase accordingly. However, our model suggests a discernible decrease in the flux of Hg(0) uptake by vegetation in these areas (Fig. 1), reflecting the differences in CO₂ and Hg element during plant physiological processes especially those related to water dynamics in terrestrial ecosystems (Wang et al., 2021).”

Reviewer #2 (Remarks to the Author):

I liked that the authors of this modeling paper were very honest about the limitations of their model calling this effort a diagnostic exercise. They have provided a nice summary of how changes in CO₂ concentrations and climate change could impact Hg uptake. I feel they were very honest about the limitations of this study.

Response: Thank you very much for your positive and constructive feedback on our manuscript. We are grateful for your recognition of our research findings and our transparency of the model limitations. Please refer to our response to specific comments as follows:

1. Some limitations not addressed clearly include impacts of deforestation that is only mentioned briefly in the implications section.

Response: Thank you for bringing this point up. In this study, our focus is on the effect of climate change on vegetation Hg(0) uptake. We haven't considered the effects of land use and land cover (LULCC), as described in the Methods section of the manuscript:

Lines 481-483: "To eliminate the impact of land use and land cover (LULCC) changes, as well as aerosols, we preprocessed the vegetation patterns and aerosol deposition in all simulation scenarios using the specified file provided by CLM5."

Your suggestion is very valuable, and in our future research, we plan to explore the interactive effects of LULCC and climate change, which could be an intriguing study. The added sentences in the revised manuscript are as follows:

Lines 386-391: "We did not consider the impacts LULCC in this study. Under global warming, vegetative succession following melting and increased precipitation intensity is likely to lead to an increase in vegetative biomass and, consequently, an increase in Hg(0) uptake by vegetation (Wang et al., 2020a). Indeed, the interactive effects of climate change combined with changes in LULCC worth further examination."

Lines 86-87: "...and remove the effects of land use and land cover change (LULCC) and aerosols (see Materials and Methods)."

2. They note that plants physiologically could adapt to higher concentrations. Is there any work to indicate how plant behavior could change?

Response: Thank you for bringing this point up. It is reported that under the influence of long-term eCO₂ conditions, stomatal density and size will decrease, which are attributed to the guard cells and mesophyll tissues that mediate stomatal movements. The improved sentences have been added to the revised manuscript as follows:

Lines 233-239 : "Intriguingly, the sensitivity of stomata to eCO₂ diminishes gradually under the influence of long-term eCO₂ conditions. This occurs because guard cells and mesophyll tissues, which mediate stomatal movements, lead to decreases in stomatal aperture and size, culminating in physiological adaptation to higher concentrations (Engineer et al., 2016; Liang et al., 2023). Consequently, this results in a less pronounced decline from the SSP3-7.0 to SSP5-8.5 scenarios (Fig. 2)."

3. They are clear that the experimental work was done with seedlings, and there will be differences associated with different plant species.

Response: Thanks for your good suggestion. We have added and improved the following sentences as follows:

Lines 285- 288: “Additionally, there will be differences associated with different plant species. Thus, we suggest that future research should focus on this aspect, aiming to bridge the knowledge gap by including experiments across various growth stages and more species.”

4. Line 287 it has been demonstrated in multiple studies that Hg does not enter the plants via the roots and that roots act as a barrier accumulating soil Hg

Response: Thank you for pointing this out. As you mentioned, Hg is hard to enters the plants via the root, as most previous studies have shown (Arnold et al., 2018; Chiarantini et al., 2016; Siwik et al., 2010). Regarding this, we have made the following supplementary explanation in the revised manuscript:

Lines 327-336: “Our model also did not consider the absorption of Hg from underground root systems and root secretions (Keuper et al., 2020). Indeed, Hg is hard to enter the plants via the root, as most previous studies have shown (Arnold et al., 2018; Chiarantini et al., 2016; Siwik et al., 2010). Meanwhile, our model does not account for the translocation of Hg among plant tissue organs. A recent study suggests that a significant proportion of Hg in roots may originate from absorption by leaves and subsequent translocation, with an estimation of up to 300 Mg yr⁻¹ of atmospheric Hg⁰ stored in roots (Zhou et al., 2021), but the specific migration and distribution mechanisms are still unclear. These processes also have a potential influence on the amount of Hg(0) uptake by the vegetation, and could be incorporated in our model when more data is available.

5. Line 320 I would change this to read “bypass sequestration of Hgo by plants and deposition of foliar Hg to the soil causing increasing concentrations in the atmosphere, where it could be converted to HgII that is deposited to ecosystems, where it can subsequently be methylated.

Gaseous elemental Hg is the form taken up by foliage. HgII is not taken up see Arnold et al. (2018)

Response: Thank you very much for the valuable suggestion. We have rewritten the sentences as advised, as follows:

Lines 370-373: With climate change, the bypassing of atmospheric Hg(0) sequestration by plants and the deposition of foliar Hg to the soil lead to increasing concentrations in the atmosphere. This Hg can then be converted to HgII, which is deposited in aquatic ecosystems and can subsequently be methylated (Arnold et al., 2018; Feinberg et al., 2022; Zhou et al., 2021).

Some general housekeeping

6. Line 28 which should be that.

Response: Revised as suggested

7. Line 48 previous research has

Response: Revised as suggested

8. Line 84 and throughout the text what was done should be described in past tense. Here indicated should be used. Line 89 were instead of are.

Response: Thanks for your suggestions. We have thoroughly checked and corrected all the tenses that needed to be changed, and the changes have been highlighted in revised manuscript.

9. References for Supplemental Information are not in alphabetical order and

Arnold J, Gustin MS, Weisberg PJ. Evidence for Nonstomatal Uptake of Hg by Aspen and

Translocation of Hg from Foliage to Tree Rings in Austrian Pine. *Environmental Science & Technology* 2018; 52: 1174-1182.

Response: We have now revised all our references to conform to the format of *Nature Communications*.

Reviewer #3 (Remarks to the Author):

The manuscript addresses a crucial topic, given the pivotal role of plants in the global cycle of toxic mercury and the current and anticipated changes in CO₂ emissions and budget. Its significance is heightened by the imperative to effectively implement the UN Minamata Convention on Mercury. Despite numerous studies on the intricate impact pathways of climate change on vegetation Hg(0) uptake, the cascading effects of climate change-related factors on the terrestrial Hg cycle, particularly the uptake of Hg(0) by vegetation, remain unclear.

The study employs a compelling combination of the Community Land Model - Hg (CLM5-Hg) and the Community Earth System Model (CESM). Although the novel method has been previously applied by the authors (Yuan et al., 2023), the current manuscript utilizes a distinct approach, yielding entirely new results. The presented findings are highly valuable and align with the global trend of research on pollutant cycling and its connections to climate change. The publication is meticulously and thoughtfully prepared, and its content is certain to capture readers' interest.

The introduction section is succinct yet effectively articulates the rationale for the research topic. The results and discussion section is well-organized, comprising four subsections that sequentially address changes in Hg uptake by vegetation, the coupling of terrestrial Hg and CO₂ uptake, model uncertainties, and potential implications. In this section, my comments primarily relate to the part associated with model uncertainty. However, they do not question the substantive value of the paper. My comments are driven more by scientific curiosity in this context. A separate section constitutes the Materials and Methods part, in which the authors provide a detailed description of the CLM5-Hg model. This model simulates the migration, transformation, accumulation, and emission processes of Hg in terrestrial ecosystems. It encompasses both stomatal and non-stomatal uptake in leaves, throughfall, soil formation and leaching, photo-reduction, microbial decomposition, thermal evaporation, and emissions from wildfires. In my assessment, the model's components allow for a comprehensive tracking of the Hg cycle involving land plants. Particularly interesting is the application of distinguishing numerous plant functional types (PFTs) characterized by different leaf, stem, and root properties. This approach has been successfully tested against observational field data in a previous study by the authors (Yuan et al., 2023). The data used regarding simulated atmospheric Hg(0) concentrations, as well as the dry and wet deposition fluxes of Hg(II), do not raise concerns, according to Zhang & Zhang (2022). Similarly, the utilization of the global vegetation Hg(0) flux dataset (Feinberg et al., 2022; Yuan et al., 2023), as well as CO₂ dataset (Millhollen et al., 2006; Sañudo-Wilhelmy et al., 2008; Stamenkovic and Gustin, 2009; Duval et al., 2011; Demers et al., 2013; Tang et al., 2021), is adequate. The uncertainty analysis appears to have been conducted thoroughly and comprehensively in my opinion.

In conclusion, the manuscript is presented in a clear and generally grammatically correct manner. However, I advise authors to carefully check through the text to ensure consistency in verb tenses, especially when discussing model results or future research directions.

Below, you will find a list of my suggestions.

Response: Thank you very much for your multifaceted recognition of our study, as well as for the detailed and thoughtful review. Your insightful comments and suggestions are invaluable to us.

Please refer to our responses to the specific comments as follows:

Introduction

1. Lines 36-37: Consider adding information on the adverse effects of Hg on the ecosystem.

Response: Thank you very much for the valuable suggestion. We have added the following sentence in revised manuscript.

Lines 38-39: “Additionally, it endangers ecosystems by bioaccumulating in food chains, affecting biodiversity and disrupting ecological balances (Eagles-Smith et al., 2016).”

2. Lines 39-42: Consider adding information that long-living vegetation can store not only present-day Hg but also Hg emitted into the atmosphere decades ago

Response: Thank you very much for the valuable suggestion. We have added this sentence you advised in the revised manuscript as follows:

Lines 42-45: “Vegetation within terrestrial ecosystems can absorb large amounts of atmospheric gaseous elementary Hg [Hg(0)] (2200-3600 Mg yr⁻¹), acting as a major sink for the atmosphere in the present-day Hg cycles (Obrist et al., 2021; Zhou et al., 2021). Long-living vegetation not only stores present-day Hg but also Hg emitted into the atmosphere decades ago (Bargagli, 2016).”

3. Line 42: Change "warming temperatures" to "rising temperatures". The same applies to line 51

Response: Thanks for the suggestions, we fixed it in the revised manuscript.

4. Line 45: Add a comma after "vegetation"

Response: Revised as suggested.

5. Lines 51-52: The symbol "Hg" has already been explained at the beginning of the Introduction.

Response: Revised as suggested.

6. Line 60: Correct the citation for "Zhang et al., 2016"

Response: Revised as suggested.

7. Line 74: What is the source of pre-industrial data? Also, correct "c.a" to "ca."

Response: Revised as suggested.

8. Line 82: Change to "Materials and Methods"

Response: Revised as suggested.

Results and Discussion

9. Line 86: Consider specifying what kind of scenarios you are referring to, in this case “business-as-usual”. The same applies to line 188

Response: Thanks for the suggestions. “SSP5-8.5 scenario” has been revised to the “SSP5-8.5 (‘business-as-usual’) scenario” in the mentioned place.

10. Lines 106-107: Consider adding “2°C scenario” and “regional rivalry”. The same applies to line 213

Response: Thanks for bringing it up. We have added them in the corresponding place.

11. Line 138: Provide a clear understanding of what the percentages represent, e.g., a change in Hg uptake or a decrease in foliage Hg levels

Response: Thank you very much for the reviewer's suggestion. We have addressed it as follows:

Lines 155-158: “The meta-analysis revealed a significant decrease in vegetation Hg levels as a result of eCO₂, showing an average decrease in foliage Hg levels or Hg uptake of 5.87% per 100 ppm increase (95%CI, -6.5% to -5.3%).”

12. Line 179: Change to "Decoupled CO₂ and Hg" since you are referring to CO₂ specifically rather than total carbon. The same applies to lines 232, 305, and 333

Response: Thank you very much for the valuable suggestion. We fixed it in the revised manuscript.

13. Lines 189-192: Consider providing a concise statement that summarizes the key finding before

delving into the specific details. This can help guide the reader's understanding

Response: Thank you very much for the valuable suggestion. We have added the following sentence:
Lines 206-208: “Our analysis identified that the sunlit stomatal conductance emerges as the key driving factor influencing the observed reduction in Hg(0) uptake.”

14. Lines 204-207: Consider breaking down this sentence into shorter ones for better clarity. The same applies to lines 208-212

Response: Thank you very much for the valuable suggestion. We have rewritten the following sentences:

Lines 223-226: “This results in reduced stomatal conductance (g_s) with eCO_2 (Supplementary Fig. 13), leading to decreased transpiration (Fig. 4d). This process induces increased soil water storage, enhancing water use efficiency (Supplementary Fig. 14).”

Lines 227-231: “Stomatal conductance is directly related to vegetation uptake of Hg(0) in our model (equations (1)-(4), for details refer to Methods). This relationship explains the gradual reduction in vegetation uptake of Hg(0). We observe this reduction from the pre-industrial and present-day periods to the SSP1-2.6 scenario (Supplementary Fig. 12).”

15. Lines 217-220: Consider providing a brief transition sentence. Also, please explain “CO₂ fertilization effect”

Response: We appreciate the reviewer for pointing this point out. We added and improved the following sentences in the revised manuscript.

Lines 240-241: “A tight coupling between carbon and Hg in terrestrial ecosystems has been observed as a paradigm over the past two decades (Smith-Downey et al., 2010; Wang et al., 2020a).”

Lines 241-244: “Conventionally, it has been postulated that rising atmospheric CO₂ levels would increase vegetation's photosynthesis rate, leading to the beneficial impact on plant growth, known as the CO₂ fertilization effect (Liu et al., 2019). This effect is believed to enhance the concurrent absorption of both CO₂ and Hg(0), as suggested by Jiskra et al. (2018), Lawrence et al. (2019), Obrist (2007), and Schaefer et al. (2020).”

16. Lines 225-227: Please clarify whether you mean a decoupling in trends or a decoupling in the magnitude of the trends

Response: Thank you for pointing this out. What we indeed meant is a decoupling in trends. We have clarified this in the last paragraph of the “Decoupled CO₂ and Hg” section in the revised manuscript:

Lines 252-263: “The CLM5 model projected an increased greening of vegetation in many regions in the 21st century resulting from eCO_2 (a.k.a. fertilization effect), evidenced by the increased leaf area index (LAI) in the northern mid-to-low latitudes and certain regions of the Southern Hemisphere (Fig. 4c). ...Therefore, under climate change, the increase in vegetation LAI may only represent an increase in leaf density or even stomatal numbers, but stomatal conductance may not necessarily increase accordingly. However, our model suggests a discernible decrease in the flux of Hg(0) uptake by vegetation in these areas (Fig. 1), reflecting the differences in CO₂ and Hg element during plant physiological processes especially those related to water dynamics in terrestrial ecosystems (Wang et al., 2021)”

17. Lines 247-249: Please rewrite this sentence for better clarity, e.g., "This occurs in part because the anomaly forcing method assumes that future changes (anomalies) can overlay present-day variability."

Response: Thank you very much for the valuable suggestion. We have rewritten this sentence in the

lines 275-277 of the revised manuscript as you advised.

18. Lines 252-253: Consider providing a brief transition sentence, e.g., "Additionally, it's important to note that some data sources in our meta-analysis originate from seedling experiments."

Response: Thank you very much for the valuable suggestion. We refined this sentence as you advised.

Lines 279-280: "Additionally, it is important to note that some data sources in our meta-analysis originate from seedling experiments."

19. Lines 271-274: Please rewrite this sentence for better clarity. Also, please be consistent in terminology used, either "Medlyn" or "medlyn"

Response: Thank you for pointing this out. We have rewritten the sentence as follows and standardized the terminology to 'Medlyn'.

Lines 300-304: "In the Medlyn model within CLM5, the Medlyn slope, denoted as "g1," plays a crucial role in controlling how stomata respond to CO₂ levels. It does this by determining the extent to which stomata open, based on the assimilation capacity, CO₂ concentration, and vapor pressure deficit (VPD).

20. Line 283-285: The statement lacks specificity. Please provide more details on the specific uncertainties in the model representation of land-atmosphere Hg exchange and its implications for future predictions

Response: Thank you for pointing this out. We clarified this point as follows:

Lines 313-321: "There are still considerable uncertainties regarding the model representation of the land-atmosphere exchange of Hg at present-day, which serves as a baseline for our prediction for the future. The atmospheric Hg concentrations and deposition are specified as a boundary condition, not yet dynamically modeled in a two-way coupled fashion. **The feedback between land Hg emissions and their atmospheric abundance and subsequent deposition onto the land are also not considered. Although our current framework can well diagnose the direct impact of changing climate on these exchange fluxes, an online land-atmosphere coupled model will be needed to reveal a more comprehensive and accurate changes in global Hg budget in future works.**"

21. Lines 285-287: Please clarify the significance of these processes in the context of your study. How do they contribute to the uncertainties mentioned earlier?

Response: Thank you for pointing this out. We have added the corresponding explanations to the revised manuscript as follows:

Lines 322-327: "Current isotopic evidence indicates that the photoreduction process is related to the re-emission of Hg(0) by vegetation leaves, with this re-emission ratio reaching nearly 30% in subtropical forest areas (Yuan et al., 2019). However, for the majority of other regions worldwide, we lack sufficient observational data to make estimates. In our model, we have only used median values as the reduction parameter. Therefore, in future research, we need to utilize more measured data to refine our parameterization scheme."

22. Lines 287-288: Please explain the rationale behind excluding the absorption of Hg from underground root systems and root secretions. This omission could have implications for the overall accuracy of your model

Response: Thank you for pointing this out. We have added the corresponding explanations to the revised manuscript as follows:

Lines 330-336: “Meanwhile, our model did not account for the translocation of Hg among plant tissue organs. A recent study suggests that a significant proportion of Hg in roots may originate from absorption by leaves and subsequent translocation, with an estimation of up to 300 Mg yr⁻¹ of atmospheric Hg⁰ stored in roots (Zhou et al., 2021), but the specific migration and distribution mechanisms are still unclear. These processes also have a potential influence on the amount of Hg(0) uptake by the vegetation, and could be incorporated in our model when more data is available.”

23. Line 288-289: Consider elaborating on why anthropogenic and legacy Hg emissions remain unchanged. Provide justification or discuss potential impacts on the study's outcomes.

Response: Thank you for your good suggestion. Because in our study, we focus primarily on the effect of climate change on the vegetation Hg(0) uptake, we have elaborated on this in the 'Experimental Design' section of our original manuscript.

Lines 682-685 of original manuscript: “This methodological choice was made to isolate and underscore the influence of climatic variables on Hg dynamics. By controlling for anthropogenic inputs, we aim to provide a clear assessment of how climate change alone can affect the biogeochemical cycling of Hg.”

Lines 315-321 of revised MS: “The atmospheric Hg concentrations and deposition are specified as a boundary condition, not yet dynamically modeled in a two-way coupled fashion. The feedback between land Hg emissions and their atmospheric abundance and subsequent deposition onto the land are also not considered. Although our current framework can well diagnose the direct impact of changing climate on these exchange fluxes, an online land-atmosphere coupled model will be needed to reveal a more comprehensive and accurate changes in global Hg budget in future works.”

Lines 352-353 of revised MS: “The model should be interpreted as a diagnostic tool designed to unveil the influence of individual factors.”

24. Lines 317-319: Consider rephrasing the beginning for smoother flow, e.g., "Our findings reveal a suppression of atmospheric Hg(0) uptake by plants across most regions in the 21st century due to reduced stomatal conductance in vegetation caused by increased CO₂."

Response: Thank you very much for the valuable suggestion. We have rewritten as your advised in the revised manuscript.

25. Lines 321-325: Consider breaking down this sentence into shorter ones for better clarity

Response: Thank you very much for the valuable suggestion. We have rewritten the following sentences:

Lines 373-377: “Furthermore, these inorganic Hg compounds are transformed into methylmercury by microbes. This process leads to the enrichment of methylmercury in riverine and marine food chains. As a result, a substantial threat to human health arises through the consumption of inland aquatic animals and seafood, including commercial fish.”

26. Lines 330-335: Consider rephrasing to enhance clarity, e.g., "The terrestrial ecosystem, recognized as a significant Hg sink, may face disruptions under future climate change scenarios, particularly with rising atmospheric CO₂ concentrations. It becomes crucial to comprehensively consider the tight coupling among Hg, CO₂, and water cycles when assessing the effectiveness of the Minamata Convention within the context of climate change."

Response: Thank you very much for the valuable suggestion. We rephrased this paragraph as you advised in the revised manuscript.

Lines 391-395: “The terrestrial ecosystem, recognized as a significant Hg sink, may face disruptions under future climate change scenarios, particularly with rising atmospheric CO₂ concentrations

(Daniel et al., 2021; Zhang et al., 2023). Therefore, it becomes crucial to comprehensively consider the tight coupling among Hg, CO₂, and water cycles when assessing the effectiveness of the Minamata Convention within the context of climate change.”

Figures

27. Figure 1: Please add in the caption the unit of time (yr⁻¹) for the global total values of Hg(0) uptake. In line 101 change to "Materials and Methods"

Response: Thanks for your suggestions. We fixed it in the revised manuscript.

28. Figure 2: Please add explanations to the scenario numbers in the legend.

Response: Thank you for pointing this out. We have added the following sentences to the legend of Figure 2 in the revised manuscript

Lines 135-140: “SSP1-2.6 represents the lowest scenario, termed the “2°C scenario,” which aims for a sustainable future. SSP3-7.0 represents a moderate scenario, described as a medium-high reference scenario within the socio-economic context of “regional rivalry.” SSP5-8.5 represents the highest scenario, also known as ‘business-as-usual,’ considered the worst-case scenario in a high fossil fuel-intensive world.

29. Figure 3: What does the purple triangle on Figure 3a mean?

Response: The purple triangle in Figure 3a represents the "Δ" in "Δ CO₂ levels". Since it appears for the first time, we have added "Δ CO₂=" in front of the numbers outside the parentheses to clarify its meaning.

30. Figure 4: It seems there is an error in the color scale legend of figures 4a and 4c. Now it is 0 - 3 - 2 - 1 instead of 0 - 1 - 2 - 3

Response: Thank you for pointing this out. We fixed it in the revised MS.

Materials and Methods

31. Lines 613-619: From what does the division of the soil pool into specific layers in the model result? Has this approach been applied before, and if so, could you please provide references?

Response: Thank you for pointing this out. The rationale for layering is primarily based on the hydrologically and biogeochemically active differences at various soil depths (Lawrence et al., 2019). Additionally, the decomposition process of soil Hg is tied to the soil carbon pool, assuming that the Hg in the soil binds with soil carbon pools of different ages (Smith-Downey et al., 2010). The transformation between different Hg pools and the microbial transformation rate are characterized using the conversion and respiration rates of the soil carbon pool. Schaefer et al. (2020) have applied this method, although they did not consider as many soil layers due to the limited depth of soil simulated in their study. We have added this explanation in the corresponding place of the revised SI:

Lines 48-75 (SI): “The decomposition process of soil Hg is tied to the soil carbon pool, assuming that the Hg in the soil binds with soil carbon pools of different ages (Smith-Downey et al., 2010). The transformation between different Hg pools and the microbial transformation rate are characterized using the conversion and respiration rates of the soil carbon pool in every layer... This scheme features base decomposition rates that vary with depth and are modified by soil temperature, water, and oxygen limitations. It also includes vertical mixing of soil carbon and nitrogen due to bioturbation, cryoturbation, and diffusion (Lawrence et al., 2019).”

32. Lines 630-631: One of the model assumptions is that anthropogenic mercury (Hg) will persist at the current level (Zhang et al., 2016). However, some estimations indicate that global anthropogenic emissions of Hg will increase in the forthcoming decades under the current legislative

scenario, e.g. Rafaj et al. (2013, <http://dx.doi.org/10.1016/j.atmosenv.2013.06.042>), Pacyna et al. (2016, <https://doi.org/10.5194/acp-16-12495-2016>), Brocza et al. (2024, <https://doi.org/10.5194/egusphere-2024-41>). Even if this rise won't be substantial compared to the present-day level, I believe it would be worthwhile to consider this aspect somewhere in the manuscript.

Response: Thank you very much for your excellent suggestion. We have taken your advice into consideration and incorporated it into the 'Implications' section of the revised manuscript:

Lines 381-385: "Furthermore, although the impact of anthropogenic source emissions is not the focus of this study, some estimations indicate that global anthropogenic emissions of Hg will increase in the forthcoming decades under the current legislative scenario (Brocza et al., 2024; Pacyna et al., 2016; Rafaj et al., 2013). Therefore, under future climate change scenarios, it is possible that there will be a greater threat to human health."

REVIEWER COMMENTS

Reviewer #1 (Remarks to the Author):

Review for “Decoupling of CO₂ and Hg uptake by global vegetation in the 21st century” by Yuan et. al. (NCOMMS-24-07636A).

The authors did an excellent job of addressing the technical concerns raised during the original review. The revised manuscript is significantly improved from the original manuscript. I recommend a few minor revisions before consideration of publication:

1. The hypothesis of this work “We hypothesize that future climate change will increase the global atmospheric Hg(0) deposition” has been suggested by quite a few other studies in addition to Wang 2020 and Wu et al. 2012, e.g., Wu et al. 2023 (DOI: 10.1021/acs.est.3c03107), Wang et al. 2022 (DOI: 10.1080/10643389.2021.1961505), and therefore does not reflect the innovation of the work. It is recommended that the authors consider a more specific hypothesis that strengthens the novelty of the study.
2. One of the primary conclusions of this work is the “decoupling between the trends of CO₂ assimilation and Hg(0) uptake by vegetation.” This conclusion appears to be purely based on the increased stomatal resistance (or increased stomatal conductance) of foliage, proposed by earlier studies referenced in the manuscript. Although this proposed pathway is plausible, the detailed mechanism of vegetative uptake and subsequent assimilation in biomass has not been substantiated with direct measurement or experimental data. A statement that the “decoupling” will occur would be an overstatement given the weight of available evidence. It is recommended that the tone of the “decoupling process” reflect the current understanding of the vegetative uptake of Hg(0).
3. The modeling processors implemented by Smith-Downey et al. (2010) in GTMM have been outdated. In particular, the model did not consider the reduction processes in soil proposed recently and the radiation transfer in soil that contributes to the reemission. It is suspected that the net Hg sink calculated by the original CLM5-Hg model (Yuan et al. 2023, 2,576 Mg/yr) was a result of the weaker reduction parameterized in the model. Additional discussion in this regard is warranted.
4. The CLM5-Hg model only considers the exchanges in terrestrial ecosystems. Based on Yuan et al. (2023), the geogenic source was not considered. Also, the contribution of anthropogenic releases and the air-ocean exchange of Hg is not included. In a way, the air-land exchange alone does not close the global mass balance. How would the model results obtained from a constrained (i.e., only air-land exchange is considered) modeling system be related to the global Hg cycle should be discussed further.

Reviewer #2 (Remarks to the Author):

My comments have been addressed adequately.

Reviewer #3 (Remarks to the Author):

I wish to extend my gratitude for the thorough revisions made by the Authors to the manuscript titled "Decoupling of CO₂ and Hg uptake by global vegetation in the 21st century" (NCOMMS-24-07636A).

The Authors have conscientiously addressed all the remarks provided in my review. These efforts have resulted in significant improvements to the manuscript. Their attention to detail and responsiveness have notably enhanced its quality, leaving no further suggestions from my end.

In my opinion, the manuscript is now suitable for publication in its current form.

Reviewer #1

The authors did an excellent job of addressing the technical concerns raised during the original review. The revised manuscript is significantly improved from the original manuscript. I recommend a few minor revisions before consideration of publication:

1. The hypothesis of this work “We hypothesize that future climate change will increase the global atmospheric Hg(0) deposition” has been suggested by quite a few other studies in addition to Wang 2020 and Wu et al. 2012, e.g., Wu et al. 2023 (DOI: 10.1021/acs.est.3c03107), Wang et al. 2022 (DOI: 10.1080/10643389.2021.1961505), and therefore does not reflect the innovation of the work. It is recommended that the authors consider a more specific hypothesis that strengthens the novelty of the study.

Response: Thank you for pointing out this issue. We rewrote the sentence as follows:

Lines 63-66: “We hypothesize that the climate will influence global vegetation Hg(0) uptake by altering the plant physiology such as the stomatal activities, with CO₂ and other meteorological factors as important driver factors.”

2. One of the primary conclusions of this work is the “decoupling between the trends of CO₂ assimilation and Hg(0) uptake by vegetation.” This conclusion appears to be purely based on the increased stomatal resistance (or increased stomatal conductance) of foliage, proposed by earlier studies referenced in the manuscript. Although this proposed pathway is plausible, the detailed mechanism of vegetative uptake and subsequent assimilation in biomass has not been substantiated with direct measurement or experimental data. A statement that the “decoupling” will occur would be an overstatement given the weight of available evidence. It is recommended that the tone of the “decoupling process” reflect the current understanding of the vegetative uptake of Hg(0).

Response: Thank you for bringing this point up. We have revised the title and relevant sentences as follows:

Title: “**Potential** decoupling of CO₂ and Hg uptake **process** by global vegetation in the 21st century”. The word “**Potential**” indicates significant uncertainty of our results, while the word “**process**” limits the scope of our results: only a diagnosis of the uptake process, not a comprehensive prediction.

Lines 30-32: “We find a **potential** decoupling between the trends of CO₂ assimilation and Hg(0) uptake process by vegetation in the 21st century, caused by the decreased stomatal conductance with increasing CO₂.”

Lines 250-251: “Contrarily, we predicted a **potential** decoupling between the trends of CO₂ assimilation and Hg(0) uptake process by vegetation in the 21st century, when considering the dynamic response of vegetation physiological activities to climate change.”

3. The modeling processors implemented by Smith-Downey et al. (2010) in GTMM have been outdated. In particular, the model did not consider the reduction processes in soil proposed recently and the radiation transfer in soil that contributes to the reemission. It is suspected that the net Hg sink calculated by the original CLM5-Hg model (Yuan et al. 2023, 2,576 Mg/yr) was a result of the weaker reduction parameterized in the model. Additional discussion in this regard is warranted.

Response: Thank you very much for the valuable suggestion. Indeed, the soil-related process of the GTMM has been outdated, which is also the focus of our future model developing. Therefore, in this study, we primary focus on the vegetation-related process. We included the following discussion in the revised manuscript:

Lines 335-338: “Furthermore, the model simplified the soil Hg processes, following GTMM (Global Terrestrial Mercury Model) (Smith-Downey et al., 2010), by focusing mainly on the microbial reduction process. It did not account for other processes like the radiative transfer in soil, photo-reduction, and other abiotic reduction processes (wang et al.,2016; Fritsche et al., 2008).”

4. The CLM5-Hg model only considers the exchanges in terrestrial ecosystems. Based on Yuan et al. (2023), the geogenic source was not considered. Also, the contribution of anthropogenic releases and the air-ocean exchange of Hg is not included. In a way, the air-land exchange alone does not close the global mass balance. How would the model results obtained from a constrained (i.e., only air-land exchange is considered) modeling system be related to the global Hg cycle should be discussed further.

Response: Thanks for your valuable and constructive suggestions. In fact, the geogenic source was considered in CLM5-Hg, since the initial soil Hg concentration from Wang et al. (2019) included the fraction of geological sources. In addition, the atmospheric Hg concentration and deposition flux, which are the boundary condition for CLM5-Hg, also include the contribution of geogenic source. However, we completely agree with you that a further work combining air-land exchange, air-ocean exchange, and anthropogenic emissions is necessary to close the global mass balance. We included the following sentences in the revised manuscript:

Lines 399-406 in ‘implication section’: “From the perspective of the global Hg cycle, considering only the air-land exchange process is insufficient to achieve global mass balance. Both anthropogenic releases and the air-ocean exchange of Hg can potentially affect atmospheric Hg levels, thus influencing the air-land exchange process. Therefore, future research should aim to further incorporate time-varying anthropogenic emissions and develop a fully coupled land-atmosphere-ocean global Hg model within CESM2. This would enable a comprehensive understanding of the complete pathway of Hg from emission to deposition.”

REVIEWERS' COMMENTS

Reviewer #1 (Remarks to the Author):

The authors have addressed the technical concerns and I have no further comment. I want to congratulate the authors on a work well done.